# Capillary flow experiments for thermodynamic and kinetic characterization of protein liquid-liquid phase separation

Emil G. P. Stender[1,4], Soumik Ray [1,4], Rasmus K. Norrild [1,4], Jacob Aunstrup Larsen[1], Daniel Petersen[2], Azad Farzadfard[1], Céline Galvagnion [2], Henrik Jensen[3] & Alexander K. Buell [1✉]

Liquid-liquid phase separation or LLPS of proteins is a field of mounting importance and the value of quantitative kinetic and thermodynamic characterization of LLPS is increasingly recognized. We present a method, Capflex, which allows rapid and accurate quantification of key parameters for LLPS: Dilute phase concentration, relative droplet size distributions, and the kinetics of droplet formation and maturation into amyloid fibrils. The binding affinity between the polypeptide undergoing LLPS and LLPS-modulating compounds can also be determined. We apply Capflex to characterize the LLPS of Human DEAD-box helicase-4 and the coacervate system ssDNA/RP₃. Furthermore, we study LLPS and the aberrant liquid-to-solid phase transition of α-synuclein. We quantitatively measure the decrease in dilute phase concentration as the LLPS of α-synuclein is followed by the formation of Thioflavin-T positive amyloid aggregates. The high information content, throughput and the versatility of Capflex makes it a valuable tool for characterizing biomolecular LLPS.

[1] Department of Biotechnology and Biomedicine, Technical University of Denmark, Søltofts Plads, Building 227, 2800 Kgs. Lyngby, Denmark. [2] Department of Drug Design and Pharmacology, Faculty of Health and Medical Sciences, University of Copenhagen, Universitetsparken 2, 2100 Copenhagen, Denmark. [3] FIDA Biosystems Aps, Generatorvej 6 A+B, 2860 Søborg, Denmark. [4]These authors contributed equally: Emil G. P. Stender, Soumik Ray, Rasmus K. Norrild. ✉email: alebu@dtu.dk

Biomolecular liquid–liquid phase separation (LLPS) is the spontaneous condensation of molecules (most prominently —proteins and nucleic acids) into a highly concentrated ("dense") phase and a dilute phase[1–14]. Due to the lack of a physical membrane, phase separated, liquid-like assemblies can readily exchange biomolecules with their environment, fuse upon contact and re-dissolve upon dilution[1,4,15,16]. Within the last decade, protein LLPS has received increasing attention, due to the discovery of LLPS being a fundamental mechanism for the formation of membrane-less organelles (MLOs). Such MLOs include P-granules, stress granules, G-bodies, nucleolus, nuclear speckles, cajal bodies, proteasome foci, $CO_2$ fixing enzyme bodies and centrosomes to name a few[1,8–13,17]. LLPS is implicated in various functionalities of cells such as enrichment of cellular components for specific functions[18–25], organizational and sequestration hubs and assembly points during cellular signal transduction[26–29]. The ability of a protein to undergo LLPS under a given set of conditions greatly depends on its amino acid sequence[30]. Intrinsically disordered regions and low complexity domains can greatly promote LLPS because of their flexible conformational nature which aids in multivalent, adhesive intermolecular interactions[31–36]. Apart from their functional implications, protein LLPS can also result in liquid-to-solid transition (due to high local concentrations) which results in protein aggregation and amyloid fibril formation[13,37–43]. Sometimes, liquid-to-solid transition can aid in the survival of the host organism[44–46]. However, in most cases, the process is detrimental and is associated with various neurological disorders such as amyotrophic lateral sclerosis, Alzheimer's disease and Parkinson's disease (PD)[38,40,42,47]. Such aberrant liquid-to-solid phase transition and amyloid fibril formation is seen in FUS[38], hnRNPA1[13], TDP-43[48], Tau[40] and α-synuclein (α-Syn)[47] to name a few.

Thus, LLPS is a critical process to life which explains the great attention it is receiving both in academic as well as in industrial settings[49]. The availability of methods that allow high throughput (HTP) characterization of biomolecular LLPS in a quantitative manner as well as the effects of ligands and additives on the phase separation behavior is of great importance in this context[50]. So far, LLPS is usually characterized in vitro by differential interference contrast (DIC) and/or fluorescence microscopy, fluorescence correlation spectroscopy[51], turbidity, and absorption at 280 nm[52]. Microfluidic approaches[53–55] to characterize the phase diagram have also been proposed as well as NMR[56], quantitative phase microscopy[57] and Raman microscopy/spectroscopy[58,59] techniques. However, easily approachable HTP methods for characterizing LLPS are still lacking. Here we present a fully automated Capillary flow experiment (Capflex) method for quantitative LLPS, which is based on flow-induced dispersion analysis (FIDA)[60]. Capflex allows characterization of dilute phase concentrations, droplet size distributions, droplet formation and maturation kinetics and thermodynamics, as well as the affinity of the protein/polypeptide undergoing LLPS for various interaction partners—all in a thermally controlled environment. Furthermore, Capflex can be easily interfaced with automated fluid handling systems, such as the OpenTrons OT2.

In this study, we employ Capflex to characterize the phase separation behavior of three unrelated proteins/peptides— Ddx4n1, RP$_3$ peptide, and α-Syn. Human DEAD-box helicase-4 (Ddx4) is a protein involved in the formation of nuage in sperm and egg cells[35] and suspected of being involved in ovarian cancer[61]. The n1 domain of Ddx4 (Ddx4n1) has been shown to be responsible for fully reversible LLPS, is intrinsically disordered and does not fold upon undergoing LLPS[35,56]. The synthetic RP$_3$ peptides have previously been applied as a model for intracellular MLOs as they form coacervates (hollow condensates) with RNA as well as with ssDNA[62,63]. Importantly, we also study α-Syn, the intrinsically disordered protein primarily associated with the pathology of PD which has been shown to undergo LLPS and subsequent aberrant phase transition resulting in amyloid fibril formation[47,64,65]. Capflex allows us to quantify the thermodynamics of the transition from reversible liquid droplets into irreversible solid particles.

## Results
Capflex uses the FIDA 1 instrument, which consists of a temperature-controlled tray in a sample-handling robot holding 96-well plates that can be maintained above the cloud point temperature (Fig. 1a). The sample is then loaded into a fused silica capillary kept in a temperature-controlled chamber (e.g. below the cloud point temperature) (Fig. 1a). When injected into the capillary, the sample undergoes LLPS and arrives at the detector after a flow rate and pressure-controlled time delay, where signal spikes are recorded as each droplet passes (Fig. 1a). The experiment can also be performed with a sample that has already undergone LLPS. The signal baseline intensity level allows determination of the protein concentration in the light/dilute phase (Fig. 1a). The method currently measures fluorescence compatible with yellow fluorescent protein (YFP) or Alexa488 dyes but can also be adapted to intrinsic protein fluorescence. The detector is sufficiently sensitive that only nano molar concentrations (nM) of labeled protein is required in the solution. For most purposes, the concentration of the labeled protein corresponds to less than 1% of the total protein in the sample. It is necessary to perform a standard curve calibration to extrapolate the dilute phase concentration(s). Generally, a known set of increasing protein concentrations with a constant proportion of labeled protein can be injected into the capillary while making sure the sample is not phase separated. This can be achieved by using a concentration range lower than the critical concentration of the protein.

**Characterization of Ddx4n1 LLPS**. To test the method, we first performed a standard curve calibration with 10, 25, and 50 µM Ddx4n1 (spiked with 200 nM (0.4%) Ddx4n1-YFP) in 20 mM Tris pH 8.0, 100 mM NaCl, 5 mM TCEP at 25 °C. To note, at this temperature and concentration range, Ddx4n1 does not undergo LLPS. We observed a linear increase in the baseline fluorescence intensity with increasing concentration of Ddx4n1 (Fig. 1b, inset). Subsequently, we induced LLPS of 50 µM Ddx4n1 by adding increasing amounts of % (w/v) PEG3000 (Fig. 1c). PEG is a well-known molecular crowder and has been shown to induce LLPS in other systems such as α-Syn.[9] Immediately after adding PEG3000, we observed spikes which indicated successful phase separation of Ddx4n1 (Fig. 1c). The baseline fluorescence intensity levels correspond to the dilute phase concentration after the system has undergone LLPS. We observed that with the increase in % PEG3000, the baseline fluorescence decreased significantly, suggesting that more and more protein molecules partitioned from the soluble phase to the droplet phase (Fig. 1c, right panel). This allowed us to quantitatively measure the dilute phase concentration of Ddx4n1 phase separation both in the absence and presence of additives such as PEG3000. We also measured the cloud point temperatures of reversible Ddx4n1 LLPS in the absence of PEG and in the presence of 2, 3, and 4% (w/v) PEG3000. Our data indicated that for 50 µM Ddx4n without PEG, the cloud point is ~20 °C. However, the cloud point temperature significantly increased (more temperature-stable droplets) with increase in PEG3000 concentration (Supplementary Fig. 3). Interestingly, Ddx4n1 showed completely reversible LLPS behavior with respect to temperature (Supplementary Fig. 3). We further established that the baseline decrease in Capflex with

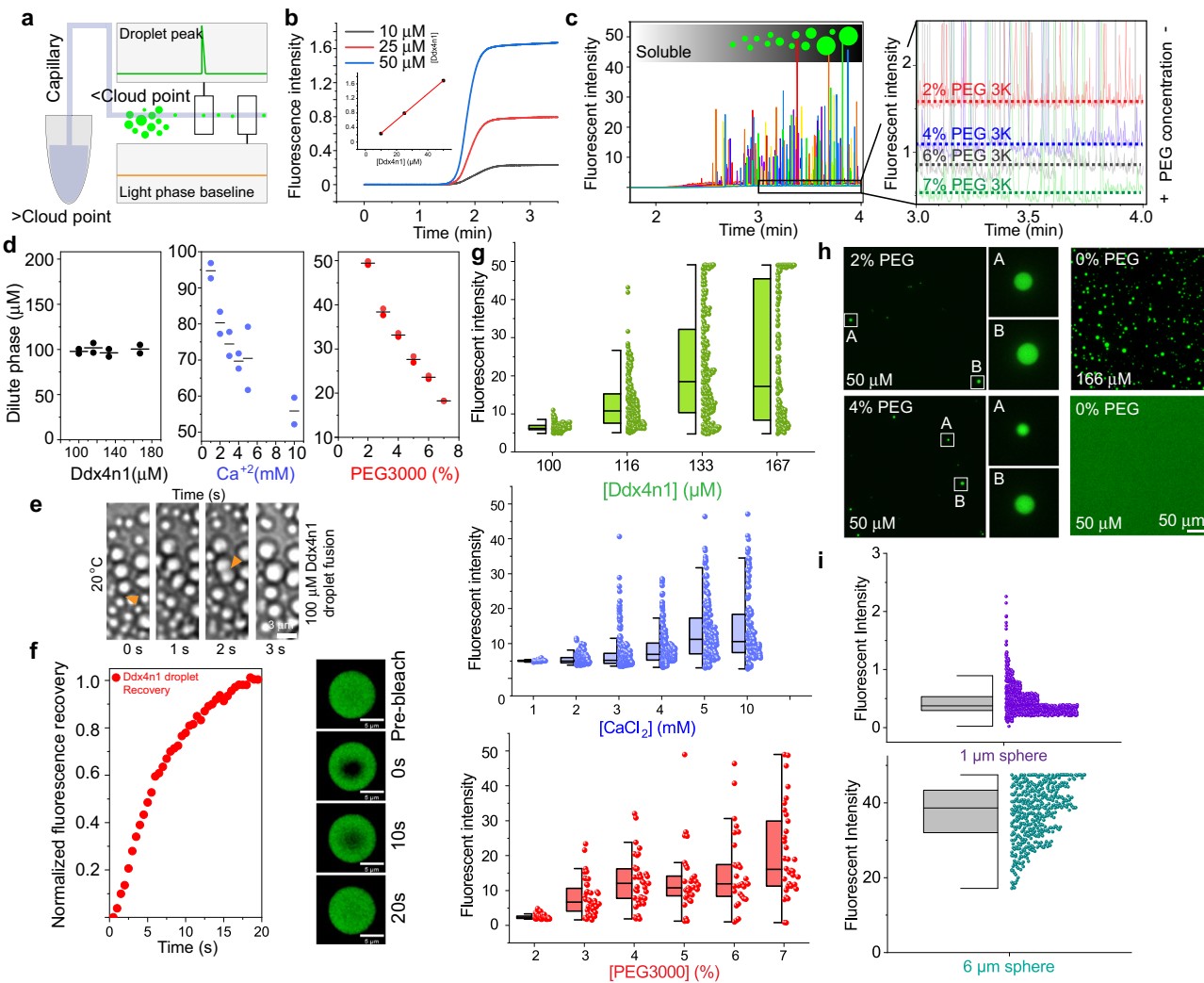

**Fig. 1 Quantitative analysis of Ddx4n1 LLPS using Capflex. a** Schematic representation of the principle behind Capflex. The samples are incubated in a thermostatted tray (>cloud point) and injected into the thermostatted capillary (<cloud point) where LLPS occurs. After flowing through the 1 m long capillary, the injected sample reaches a detector where a droplet causes a spike in the signal. The baseline between spikes corresponds to the dilute phase concentration. **b** Standard curve calibration of the baseline using 10, 20, and 50 μM soluble Ddx4n1 is shown. **c** Signal spikes are observed when 50 μM Ddx4n1 is phase separated in presence of PEG3000 below the cloud point temperature (left panel). The fluorescence baseline changes in response to LLPS inducing additive (% (w/v) PEG3000) (right panel). **d** Dilute phase concentrations for Ddx4n1 alone (left) and as a function of additives (Ca$^{+2}$ (middle) and PEG3000 (right)) at 20 °C is shown. The data represents replicate values and mean (black line) from $n = 2$ independent experiments. **e** Ddx4n1 droplet fusion events are shown as time-lapse snapshots for 3 s. The successful fusion events are marked with yellow arrowheads. $n = 2$ independent experiments. **f** (Left panel) Normalized fluorescence recovery of Ddx4n1 droplets for 20 s. (Right panel) Representative images before bleaching, just after bleaching (0 s), at 10 s and at 20 s are shown. $n = 2$ independent experiments. **g** (Upper panel) Peak intensity distribution as a function of Ddx4n1 concentration. (Middle panel) Peak intensity distribution as a function of [Ca$^{2+}$]. (Lower panel) Peak intensity distribution as a function of [PEG3000]. $n = 2$ independent experiments. **h** (Left panel) Representative fluorescence microscopy images of 50 μM Ddx4n1 with 2 and 4% (w/v) PEG3000. The insets are zoomed in on the right panel for better visualization of the droplets. (Right panel) Representative fluorescence microscopy images of 166 μM Ddx4n1 without PEG3000. Notably, 50 μM Ddx4n1 in absence of PEG3000 (non-phase separated sample) show a diffuse signal under the microscope. $n = 2$ independent experiments. **i** Peak intensity distribution of 1 μm (left panel) and 6 μm (right panel) fluorescent polystyrene spheres are shown. $n = 2$ independent experiments. Source data are provided as a Source Data file.

increasing PEG3000 concentration was solely due to an increasing degree of LLPS, as PEG3000 does not influence YFP fluorescence (Supplementary Fig. 3). Quantitative analysis of the dilute phase showed that the addition of as little as 2% (w/v) PEG3000 resulted in significant decrease in the critical concentration (from 100 to 50 μM). An increase in PEG3000 concentration resulted in a further decrease of the dilute phase concentration of Ddx4n1 LLPS (Fig. 1d).

Canonical LLPS demands that the dilute phase concentration will remain identical even though the total protein concentration is increased (given that the protein is above the critical

concentration)[4]. To check this, we used a range of Ddx4n1 concentrations (from 100 to 167 μM) and subjected the samples to LLPS at 20 °C in Capflex. Our data showed the concentrations of dilute phase does not change with total protein concentration; only the relative volume fractions of dilute and dense phase (Fig. 1d)[50]. The assumption that it is the volume fraction and not the concentration of the dense phase that increases is based on the currently accepted standard view of LLPS, where movement parallel to the concentration axis leads to changes in relative volume fraction. These Ddx4 droplets showed liquid-like behavior as confirmed by frequent fusion events (Fig. 1e and

Supplementary Movie 1) and ∼100% fluorescence recovery in fluorescence recovery after photobleaching (FRAP) measurements (Fig. 1f). This was further confirmed using Capflex by exploiting the ability of pre-formed Ddx4n1 liquid droplets to dissolve upon heating above the cloud point temperature (at 50 °C) (Supplementary Fig. 4).

Next, we utilized Capflex to investigate the effect of more biologically relevant additives. $Ca^{2+}$ is an important regulator of intracellular processes in eukaryotes[66,67]. We observed that, as the concentration of $Ca^{2+}$ is increased, the dilute phase concentration decreases in an asymptotic manner towards 65 μM (Fig. 1d). In agreement with similar observations[67], our data indicates a potential role of intracellular $Ca^{2+}$ in regulation of Ddx4n1 LLPS. This effect is unlikely to be due to an increase in ionic strength as increasing NaCl concentration increases the dilute phase concentration of Ddx4n1; i.e. decreases the driving force for LLPS[35]. To check whether the observed effect was specific for $Ca^{2+}$, we performed a control experiment with $Mg^{2+}$. We find that similar to $Ca^{+2}$, the dilute phase concentration was also substantially reduced in the presence of 10 mM $Mg^{2+}$ (Supplementary Fig. 5)—indicating that observable decrease in the dilute phase concentration is not specific for $Ca^{+2}$, but rather could be a more generic effect of bivalent cation binding of Ddx4n1 modulating its LLPS behavior[56,67,68].

We then quantified the peak (droplet spike) height distributions of only Ddx4n1 (100–167 μM) (Fig. 1g, upper panel) and 50 μM Ddx4n1 in the presence of 1–10 mM $Ca^{+2}$ (Fig. 1g, middle panel) and 2–7% (w/v) PEG3000 (Fig. 1g, lower panel). In these experiments, we noted that the median peak (droplet spikes) height increases, and the distribution widens as a function of increasing Ddx4n1 concentration; increasing concentration of $Ca^{+2}$; and increasing concentration of PEG3000 (Fig. 1g, Supplementary Table 1). Interestingly, as only the volume fraction between the two phases should change in these experiments, we hypothesized that the peak intensity distribution could reflect the overall droplet size distribution. To investigate in more detail the relationship between the spike intensity distribution and the

actual size distribution of the droplets, we measured droplet sizes of 100, 116, 133, and 167 μM of phase-separated Ddx4n1 samples using a confocal fluorescence microscope. Our observations were consistent with our Capflex results with respect to increase in droplet size and widening of the distribution as a function of total protein concentration (Supplementary Fig. 6, Supplementary Table 1). To note, in our microscopic analysis, sedimented condensates were not taken into account. However, since the Capflex experiments are performed in the laminar flow regime, there could be a bias in the distribution of peak intensities. This is because the droplets spend different time periods in the detection volume, depending on whether they are subject to the rapid flow in the center of the capillary or slower flow at the walls (Methods section). In order to quantify the magnitude of this potential bias, monodisperse polystyrene spheres with hydrodynamic diameter of 1 or 6 μm were analyzed (Fig. 1i, Supplementary Table 1). Intriguingly, the 1 μm spheres yield an intensity distribution that is very narrow and similar to the profile observed by measuring the fluorescence intensity distribution of the spheres by fluorescence microscopy (Supplementary Fig. 7, Supplementary Table 1). For the 6 μm spheres the distribution is broader and the spheres saturate the detector at 50 FU, causing the distribution to be skewed (Fig. 1i, Supplementary Table 1). As only a few of the experimentally recorded peak intensities of Ddx4n1 approach 50 FU (Fig. 1), we concluded that the relative peak intensity distribution therefore closely reflects the droplet size distribution. The highly dilute condition of labeled protein (nM range) prevents any significant contribution of inner filter effects, and therefore the peak intensity can be expected to scale with droplet volume.

**Effect of ssDNA on Ddx4n1 LLPS.** We next investigated the effect of DNA on the LLPS of Ddx4n1, as it has been reported to partition single stranded DNA (ssDNA) very strongly[35,68]. We used an ssDNA sequence previously reported to co-partition into Ddx4n1 droplets[68] (Method section) and titrated a solution of 128 μM Ddx4n1 with this ssDNA with a concentration range of ∼0–30 μM (Fig. 2a, left panel). Our data showed that the dilute

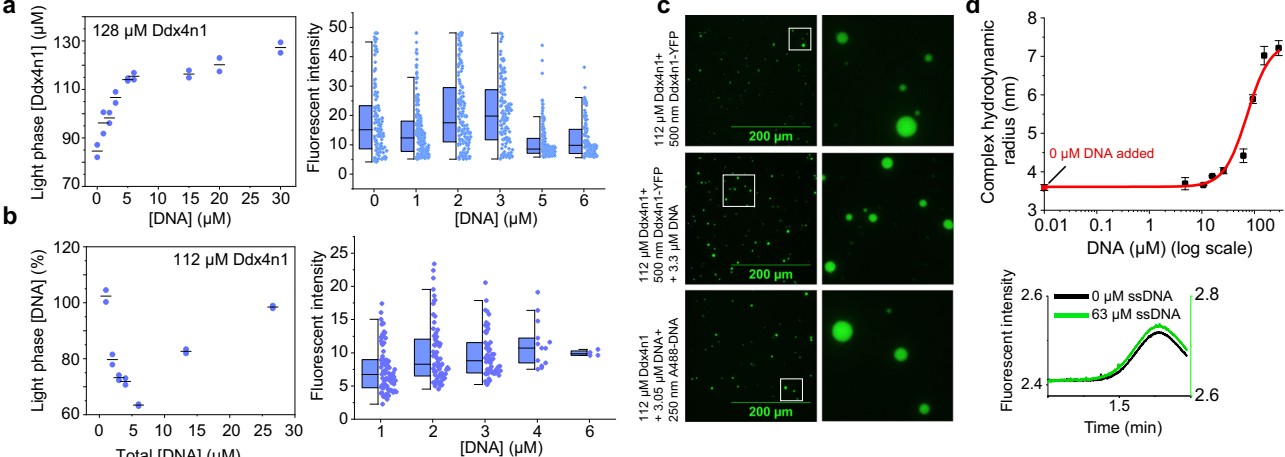

**Fig. 2 Influence of ssDNA-Ddx4n1 biding on LLPS. a** (Left panel) Dilute phase concentration of a 128 μM Ddx4n1 (spiked with Ddx4n1-YFP) as a function of added unlabeled ssDNA is plotted. The data represents replicate values and mean (black line) from n = 2 independent experiments. (Right panel) Peak intensity distribution of 128 μM Ddx4n1 as a function of unlabeled ssDNA added. n = 2 independent experiments. **b** (Left panel) dilute phase ssDNA (spiked with 5% L-DNA) concentration as a function of total ssDNA added to 112 μM unlabeled Ddx4n1 is plotted. The data represents replicate values and mean (black line) from n = 2 independent experiments. (Right panel) Peak intensity distribution of 112 μM unlabeled Ddx4n1 as a function of total ssDNA added with 0.5% spike of L-DNA. n = 2 independent experiments. **c** Representative fluorescence microscopy images of droplets formed by 112 μM Ddx4n1 with either Ddx4n1-YFP or Alexa488-DNA added as the fluorescent tracer. The insets are magnified on the right panel for better clarity. n = 2 independent experiments. **d** (Upper panel) Affinity determination between Ddx4n1-YFP and ssDNA is shown with increasing ssDNA concentration. The point with 0 μM DNA added is indicated in red. The values represent mean ± SD for n = 3 independent experiments. (Lower panel) Taylor grams with 0 (black) and 63 μM (green) ssDNA is shown for Ddx4n1. Source data are provided as a Source Data file.

phase concentration increased with ssDNA concentration (Fig. 2a, left panel) and approaches the total protein concentration asymptotically as a function of the ssDNA concentration (Fig. 2a, left panel). This indicated that ssDNA decreases the driving force of Ddx4n1 to undergo LLPS. This conclusion is further supported by the decrease of the median and width of the droplet size distributions as a function of ssDNA concentration (Fig. 2a, right panel, Supplementary Table 1). In order to confirm that the ssDNA enters the Ddx4n1 droplets we inverted the labeling scheme and utilized a spike of Alexa488 labeled ssDNA (L-DNA) in combination with unlabeled Ddx4n1. Subsequently, samples of 112 μM unlabeled Ddx4n1 was titrated with increasing concentrations of ssDNA (~1–30 μM). We observed that the ssDNA concentration in the dilute phase as a percentage of total ssDNA decreases asymptotically until ~6 μM (Fig. 2b, left panel). At 1 μM total ssDNA only small quantities of ssDNA entered the droplets as most is found in the dilute phase (Fig. 2b, left panel). However, at 2–4 μM total DNA about 30% ssDNA entered the droplets (Fig. 2b, left panel). Strikingly, at 6 μM, the majority (about 60%) of the ssDNA was found in the dilute phase (Fig. 2b, left panel). This is because at ssDNA concentrations >6 μM, the Ddx4n1 droplets started to re-dissolve back into the solution (Fig. 2b, left panel). The fact that only few droplets were observed in this condition indicates that the observed droplets are highly enriched in DNA and/or the droplets were too small to be clearly resolved as such by the instrument. The latter can be rationalized since 112 μM Ddx4n1 is close to the critical concentration for LLPS at 6 μM L-DNA (Fig. 2b, left panel and Supplementary Fig. 8). The droplets were entirely dissolved at high ssDNA concentration of 26.6 μM (Fig. 2b, left panel), where all the ssDNA was again found to be in the dilute phase. The width of the droplet size distributions as a function of ssDNA concentration substantially decreased which further supports our conclusions on the DNA-induced droplet dissolution (Fig. 2b, right panel, Supplementary Table 1). The increase in the median value of the fluorescence intensity was due to addition of increasing amounts of ssDNA (Fig. 2b, right panel).

In order to understand how such a highly sub-stoichiometric concentration of ssDNA can dissolve the droplets, we determined the relative enrichment factor of Ddx4n1 to ssDNA in the droplets. To do this, we analyzed samples with identical concentrations (112 μM Ddx4n1 and 3.3 μM ssDNA) with the only difference being whether there was a signal spike of L-DNA or Ddx4n1-YFP (Fig. 2c and Supplementary Fig. 9), i.e., in one case the DNA was labeled and in the other the protein. In this case the volume fraction of dense phase, as well as the concentrations of ssDNA and protein in the dense phase can be expected to be the same. We estimated the relative enrichment by calculating the ratio between the integral of the signal spikes and the baseline integral for both the labeling conditions and then calculated the ratio between these two ratios (Methods section). This procedure eliminated the need to know the volume fraction of dense phase. This procedure assumes that neither the fluorescence of the YFP nor of the Alexa tag is significantly different outside and inside the droplets. We obtained an enrichment factor between protein to DNA in the droplets of 1.5 ± 0.26. DNA and protein are therefore present in the dense phase at a similar stoichiometry to the overall stoichiometry, in this case a ~35 fold excess of protein[56].

Next, to delineate how a strongly sub-stoichiometric concentration of ssDNA can significantly destabilize the droplet phase of Ddx4n1, we measured the affinity of Ddx4n1-YFP for ssDNA using FIDA 1. Interactions between the positively charged arginines and the negative charge on ssDNA could potentially screen the phenylalanine-arginine interactions important for Ddx4n1 LLPS. We took advantage of the inbuilt function of the FIDA 1 instrument to measure the affinity and complex size of an indicator and analyte molecule by Taylor dispersion[69,70]. The ability to obtain interaction affinity data is an added benefit of the FIDA 1-based Capflex method. We chose to utilize Ddx4n1-YFP as fluorescence indicator as the concentration could be kept in the nM range, preventing complications stemming from LLPS, and the ssDNA as analyte. The interaction between Ddx4n1-YFP was calculated to be rather weak with a dissociation constant ($K_d$) of 50.9 ± 11.1 μM (Fig. 2d). The observed increase in size, from ~3.5 to 7 nm suggests that more than a single DNA molecule interacts with one molecule of Ddx4n1, which is why a 1:2 binding model was applied (Methods section).

**Kinetics of Ddx4n1 LLPS.** We then investigated whether our methodology could be used to monitor initial droplet formation kinetics, something that is currently difficult to measure at HTP. We used 50 μM Ddx4n1 in 20 mM Tris pH 8.0, 100 mM NaCl, 5 mM TCEP with increasing amounts of % (w/v) PEG3000 (2, 3, and 4%) to demonstrate the ability of Capflex to resolve the kinetics of droplet formation. We kept the samples initially at 55 °C above the cloud point where they are soluble and then injected into the capillary at 20 °C, below the cloud point temperature. The capillary was loaded with sample at the highest possible flow rate (3500 mbar) and thus filled within 15 s, followed by a slow injection during which peak intensity development was followed over time (Fig. 3). At the edge of the binodal line with 2% (w/v) PEG3000 and 50 μM Ddx4n1 there is a significant delay of 30 s before peaks are detected and the signal reaches a steady state after about 1.2 min at average peak intensity of 1.2 (Fig. 3a, d). As the PEG3000 concentration increases and the reaction conditions correspond to a larger driving force for LLPS, we observed no delay in LLPS and the increase in average peak intensity was much more rapid (Fig. 3b–d). For 3 and 4% (w/v) PEG3000, the maximum peak intensities reached after 1.5 min, indicating that the initial rapid rate of droplet growth slows down significantly once the droplets have reached a certain size (Fig. 3b–d).

**LLPS behavior of RP₃ peptide in the presence of ssDNA.** We also applied our Capflex approach to a small peptide system (RP₃) undergoing LLPS through coacervation to further demonstrate its versatility. RP₃ has been shown to undergo LLPS in the presence of both RNA and ssDNA which, in certain conditions, leads to coacervation of the peptide into hollow spherical assemblies[62,63]. To note, the cloud point temperature of RP₃ peptide was higher than the reachable temperature by the current FIDA 1 instrument. Therefore, in our experiments, we added ssDNA to RP₃ just prior to our measurements in the Capflex setup (Methods section). We took 184 μM RP₃ (in 10 mM Tris pH 8.0, 50 mM NaCl) and checked for its LLPS behavior in the presence of increasing concentrations of ssDNA at 20 °C. In these experiments, we used 0.17% Alexa488 labeled ssDNA as our fluorescence indicator. Our Capflex data showed minimal observable spikes (droplets) at a very low concentration of ssDNA (3.3 μM) (Fig. 4a). However, when the ssDNA concentration was increased to 13.3 and 33.3 μM, we could observe substantial droplet formation as confirmed by spikes (Fig. 4a). Strikingly, when the ssDNA concentration was further increased to 40 μM, no peaks were visible in Capflex suggesting excess of ssDNA inhibits RP₃ LLPS (Fig. 4a). The baseline fluorescence intensities steadily increased with ssDNA concentration because we used fluorescently labeled ssDNA (0.17% Alexa488 labeled ssDNA) as our indicator in the samples (Fig. 4a). Droplet size distribution analysis showed widening of the distribution with increase in ssDNA concentration from 3.3 to 33.3 μM (Supplementary Fig. 10). We

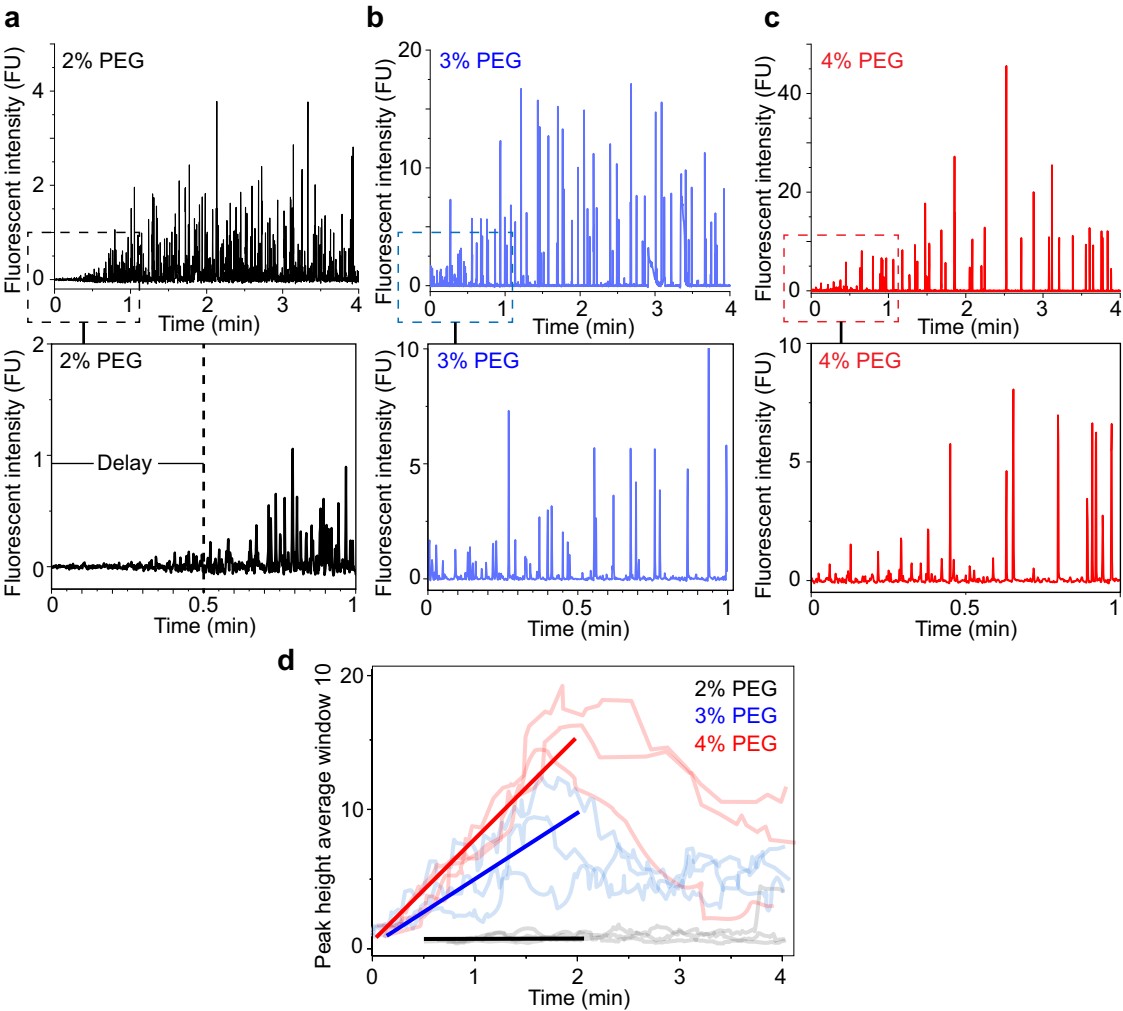

**Fig. 3 Droplet formation kinetics of 50 µM Ddx4n1 as a function of PEG3000 concentration. a** (Upper panel) Peak intensity development of 50 µM Ddx4n1 and 2% (w/v) PEG3000 for 4 min of recording time is shown. (Lower panel) Peak intensity development of 50 µM Ddx4n1 and 2% (w/v) PEG3000 from 0 to 1 min of recording time depicts the 30 s delay in the formation of droplets. **b** (Upper panel) Peak intensity development of 50 µM Ddx4n1 and 3% (w/v) PEG3000 for 4 min of recording time is shown. (Lower panel) Peak intensity development of 50 µM Ddx4n1 and 3% (w/v) PEG3000 from 0 to 1 min of recording time depicts no delay in the formation of droplets. **c** (Upper panel) Peak intensity development of 50 µM Ddx4n1 and 4% (w/v) PEG3000 for 4 min of recording time is shown. (Lower panel) Peak intensity development of 50 µM Ddx4n1 and 4% (w/v) PEG3000 from 0 to 1 min of recording time depicts no delay in the formation of droplets. However, the peaks are substantially higher under this condition compared to 2 and 3% (w/v) PEG3000. **d** Peak intensity development with an averaging window of 10 peaks of 2% PEG3000 (black), 3% PEG3000 (blue) and 4% PEG3000 (red) is shown. The solid lines are linear fits of the peak heights obtained from three independent observations for each sample (Ddx4n1 in the presence of 2, 3, and 4% (w/v) PEG3000) from 0 to 2 min. The experiments are performed three times with similar observations. Source data are provided as a Source Data file.

further confirmed this behavior with fluorescence microscopy where we analyzed three different ratios of RP$_3$ to ssDNA. At high RP$_3$/ssDNA (184 µM peptide, 10 µM ssDNA), we observed very few droplets which were also very small in size indicating that this condition is possibly at the borderline of an environment conducive to LLPS of RP$_3$ (Fig. 4b). At intermediate RP$_3$/ssDNA (46 µM peptide, 10 µM ssDNA), we observed LLPS (Fig. 4b). Importantly, the droplets formed at this condition showed a ring-like morphology with a brightly fluorescent boundary and a darker core indicating a hollow nature of the droplets (Fig. 4b, right panel). Subsequent incubation of the sample at 20 °C for 20 min resulted in substantial growth of these hollow coacervates supporting their liquid-like nature (Fig. 4b, right panel). Consistent with our Capflex data, we did not see any droplet formation under the microscope when a low RP$_3$/ssDNA ratio (46 µM peptide, 70 µM ssDNA) was used (Fig. 4b).

Next, the affinity of ssDNA for RP$_3$ was measured (similarly to the affinity of Ddx4n1 for ssDNA, Fig. 2) using the Taylor dispersion measurement functionality of the FIDA 1 instrument. We used the Alexa488 labeled ssDNA as the indicator in our affinity experiments. Additionally, the samples were prepared using the OpenTrons OT2 pipetting robot. The hydrodynamic radius of the ssDNA was measured to be 2.3 ± 0.2 nm in the unbound form, in close agreement with the expected value of 2.2 nm[71]. As the RP$_3$ peptide is titrated in, the apparent hydrodynamic radius decreases and eventually approaches 0.8 ± 0.0 nm indicative of a strong collapse of the DNA structure. This highlights the strong effect of charge neutralization of the 10 kDa negatively charged DNA by the 1 kDa positively charged RP$_3$ peptide. The difference in size of the ssDNA and RP$_3$ peptide makes it likely that more than one peptide molecule can bind to each DNA molecule. We therefore fitted a one to two binding

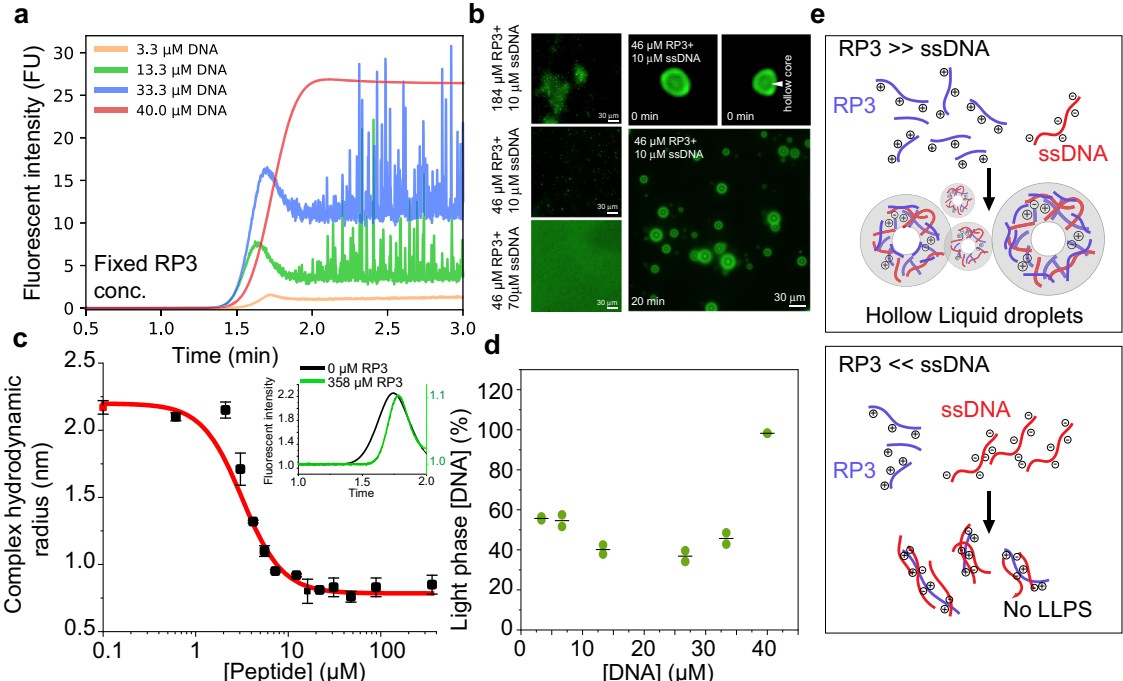

**Fig. 4 Coacervation of the arginine-rich peptide RP3 and ssDNA shows non-linear behavior as a function of DNA. a** Capflex data showing the titration of 184 μM $RP_3$ with increasing concentration of ssDNA (3.3 μM, orange; 13.3 μM, green; 33.33 μM, blue; 40 μM, red) containing a fixed ratio of Alexa488 labeled ssDNA (0.17%). As $RP_3$ does not form droplets on its own, spikes on the traces are only observed at intermediate concentrations before being dissolved at higher ssDNA concentrations. The initial overshoot of the signal under conditions where droplets are observed might arise from droplets in the fast flow in the center of the capillary, which then re-dissolved when surrounded only by buffer. **b** Representative fluorescence micrographs of $RP_3$/ssDNA coacervates in the different regimes showing the disappearance of droplets at high ssDNA: $RP_3$ ratios. In the intermediate regime, hollow coacervates were observed (right panels). These hollow coacervates grow in size after 20 min incubation at 20 °C indicating liquid-like behavior. $n = 2$ independent experiments. **c** FIDA measurements of Alexa488-ssDNA hydrodynamic radius when titrated with $RP_3$ shows a strong decrease in size upon binding, with an apparent $K_d = 5.3 \pm 0.4$ μM. The red point corresponds to 0 μM $RP_3$ and the inset shows Taylor grams with 0 (black) and 358 μM (green) $RP_3$ added. Values represent mean ± SD for $n = 3$ independent experiments. **d** Using standard curve calibration, the light phase ssDNA concentration is estimated and plotted as percentage of the total concentration of ssDNA. The data represents replicate values and mean (black line) from $n = 2$ independent experiments. **e** Schematic showing $RP_3$ droplets are only formed at intermediate ratios of ssDNA to peptide (upper panel), while an excess of ssDNA presumably saturates the positive charges on $RP_3$ preventing the formation of the network structure needed for coacervation (lower panel). Source data are provided as a Source Data file.

model to the data, as in the case of Ddx4n1 interaction with ssDNA (Fig. 4c and Supplementary Fig. 11) yielding a $K_d$ of $5.3 \pm 0.4$ μM. This result highlights the difference in interaction strength with ssDNA between the coacervation-dependent $RP_3$ peptide and Ddx4n1 (Fig. 2). We further quantified the data obtained from Capflex and found out that the amount of ssDNA in the dilute phase decreases down to ~40% at 13 μM total ssDNA, yielding an optimal mass fraction for coacervation of 0.6 (Fig. 4d). The difference with the optimal mass fraction for coacervation of $RP_3$/ssDNA of ~2 reported previously could stem from the ssDNA used here being 10 times smaller and monodisperse[62]. When the ssDNA concentration becomes high enough the droplets eventually dissolve entirely as seen from the dilute phase concentration (Fig. 4d) and in agreement with previous reports[62]. Taken together our data indicates that $RP_3$ LLPS is greatly dependent on the stoichiometric ratio of the peptide to ssDNA. When $RP_3$ is in abundance, ssDNA can induce LLPS by electrostatic interaction with the peptide. However, at very high ssDNA concentrations, complete charge neutralization occurs for $RP_3$ which is detrimental for condensation (Fig. 4e).

**Characterizing α-Syn LLPS**. In order to further test the versatile nature of Capflex in studying LLPS at HTP, we used α-Syn phase separation as our next model system. α-Syn LLPS occurs at relatively high, yet pathologically relevant protein concentrations and is being discussed as facilitating the nucleation of amyloid fibrils[64,65]. We took 100 μM of α-Syn (in 10 mM sodium phosphate buffer saline (PBS) ($Na_2HPO_4 + NaH_2PO_4$), pH 7.4, 200 mM NaCl) in the presence of 20% (w/v) PEG6000 and incubated the sample at 37 °C to induce LLPS[65]. The sample was spiked with 5 nM Alexa488-maleimide labeled α-Syn A140C (Methods section). We confirmed successful LLPS by both bright field as well as fluorescence microscopy (observations recorded at 24 h) (Fig. 5a). These droplets showed fusion upon contact and ~100% recovery in FRAP indicating their liquid-like nature (Fig. 5b, c and Supplementary Movie 2). Additionally, a drop-casted solution of phase-separated α-Syn was mixed with a drop-casted buffer solution and the solution boundary was monitored under the microscope (Fig. 5d). Interestingly, we observed, in real time, that the phase-separated droplets at the boundary gradually dissolved upon contact with the buffer solution (Fig. 5d and Supplementary Movie 3). The dilution at the boundary pushed the α-Syn below the critical concentration, rendering droplets unstable. Their rapid dissolution confirms their liquid-like, reversible nature. However, contrary to dilution, we found that an increase in solution temperature has minimal effect on α-Syn LLPS. When α-Syn LLPS samples were subjected to a thermal ramp from 15 to 60 °C, we found that the 636 nm static light scattering intensity (indicative of droplet size and number) only

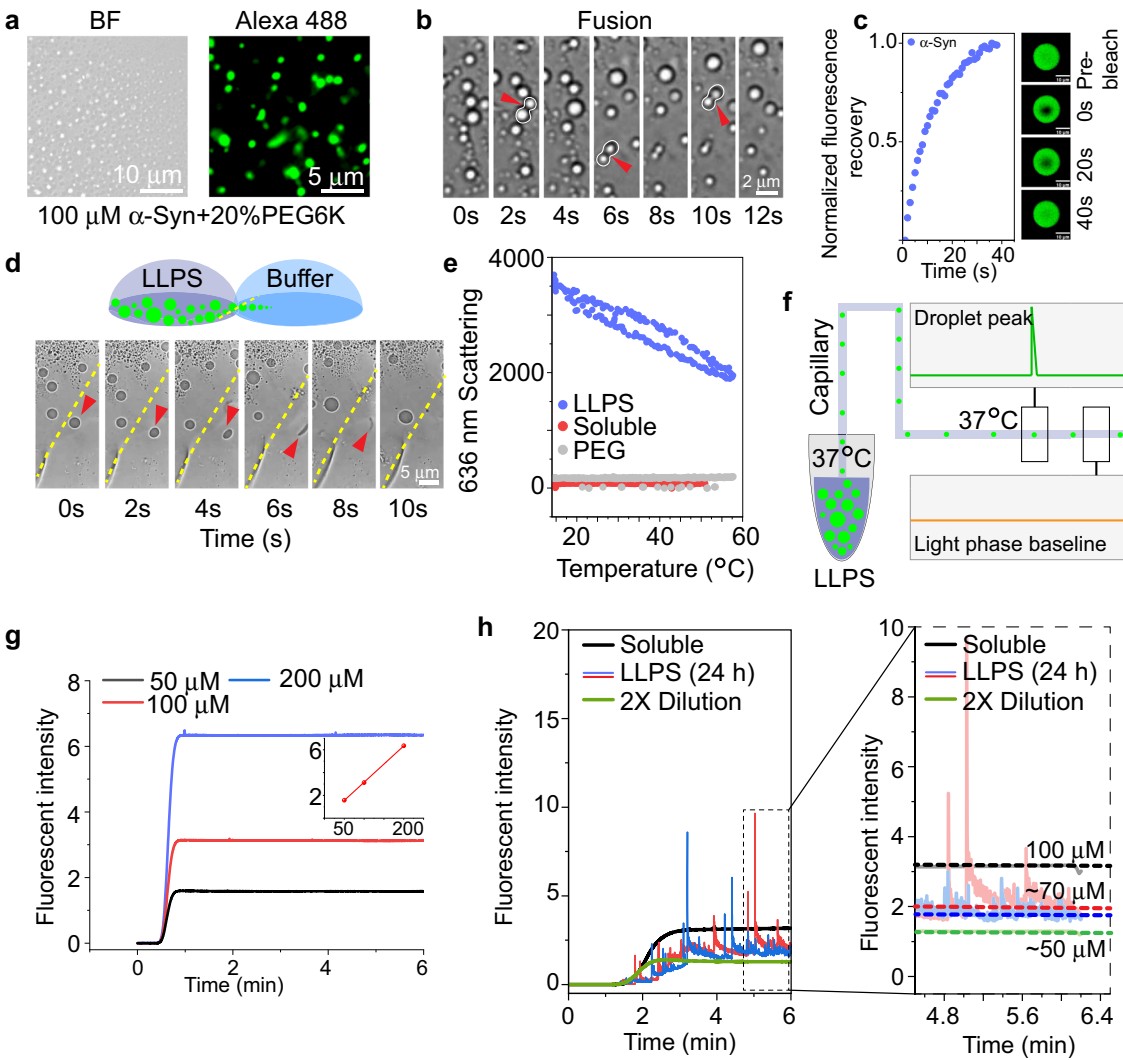

**Fig. 5 Establishing α-Syn LLPS characterization using Capflex. a** Representative bright field (left) and fluorescence (right) microscopy images of α-Syn liquid droplets are shown. $n = 2$ independent experiments. **b** Time-lapse images of α-Syn droplet fusion events are depicted. Each frame represents a 2 s time-window. The fusion events are marked with red arrows and white outlines. $n = 2$ independent experiments. **c** Normalized fluorescence recovery of α-Syn droplet is plotted for 20 s. Representative images before bleaching, just after bleaching (0 s), at 20 and at 40 s are shown. $n = 2$ independent experiments. **d** (Upper panel) Schematic depicting the solubilizing of droplets via dilution. On a glass coverslip, the LLPS sample is drop-casted. Another drop of pure buffer is casted beside the condensate containing drop in a manner so that the two drops just touch each other. When imaged with an inverted objective lens, at the border of the two solutions, the droplets show re-mixing upon dilution. Time-lapse images of remixing of α-Syn droplets are shown. Each frame represents a 2 s time-window. The border of the two solutions is marked with a yellow dotted line. The dissolving droplet is marked with a red arrow. $n = 2$ independent experiments. **e** Scattering intensity of α-Syn LLPS sample with a thermal ramp is plotted. $n = 2$ independent experiments. **f** Schematic depicting the experimental setup for Capflex when the α-Syn LLPS system is used. Here, both the sample tray and the capillary is kept at 37 °C. **g** Standard curve calibration of the baseline using 50, 100, and 200 μM soluble α-Syn (labeled with Alexa488-α-Syn) is shown. **h** Signal spikes are observed when 100 μM α-Syn is phase separated in the presence of 20% (w/v) PEG6000. The samples are observed after 24 h of incubation at 37 °C (left panel). The dilute phase concentration decreases from 100 to ~70 μM after 24 h. The dotted lines indicate the baseline fluorescence intensities (right panel). The fluorescence baseline further decreases upon dilution with buffer and the signal spikes disappear indicating dissolution of the droplets. $n = 2$ independent experiments. Source data are provided as a Source Data file.

decreased to ~50% at 60 °C—which was completely recovered when the temperature was reverted to 15 °C. (Fig. 5e) This means that the cloud point temperature of 100 μM α-Syn (in presence of 20% (w/v) PEG6000) is above 60 °C which is also consistent with previous observations[72]. Because of its temperature insensitive nature, similar to the case of the RP₃ peptide, we decided to use already phase-separated α-Syn samples in the 96-well plates in the Capflex setup at 37 °C and subsequently injected them inside the capillary (also kept at 37 °C) (Fig. 5f). We used 50, 100, and 200 μM of α-Syn (in 10 mM PBS pH 7.4, 200 mM NaCl) spiked with Alexa488 labeled α-Syn for obtaining a standard curve in

Capflex (Fig. 5g). Next, a solution of 100 μM of α-Syn spiked with 5 nM of Alexa488 labeled α-Syn A140C in the presence of 20% (w/v) PEG6000 was prepared and immediately analyzed using Capflex. The baseline fluorescence intensity showed a dilute phase concentration of 100 μM indicating no LLPS had occurred at time = 0 (Fig. 5h). Subsequently, the sample was incubated at 37 °C to induce LLPS[47]. After 24 h, Capflex identified successful phase separation with a dilute phase concentration of ~70 μM (critical LLPS concentration) indicating that 30% of the protein partitioned into droplets. Consistent with our microscopic observations, upon diluting the sample 50% (v/v) with buffer, we

observed that the signal spikes corresponding to the droplets disappeared suggesting the liquid-like, reversible behavior of the droplets at 24 h (Fig. 5h).

**Simultaneous monitoring of α-Syn LLPS and aggregation using Capflex**. α-Syn LLPS has been reported to undergo a liquid-to-solid phase transition which results in amyloid fibril formation inside the droplets[47,65,73]. We set out to study this process in a quantitative manner using Capflex by simultaneously monitoring LLPS behavior of 100 μM of α-Syn in the presence of 20% (w/v) PEG6000. We used three sets of samples to probe LLPS and the liquid-to-solid transition of α-Syn using Capflex. All the sets were prepared from one single master stock solution. (a) The first set of samples were spiked with 10 nM of Alexa488 labeled α-Syn and Capflex measurements were performed at an interval of 4 h (during 48 h). At each time-point, the sample was diluted with buffer and the reversibility (disappearance of signal spikes upon re-mixing) of the droplets was probed using Capflex (Fig. 6a, c). (b) The second set of samples were spiked with 5 μM of ThT and recorded at different time-points (for 48 h) to monitor amyloid aggregation[74,75] in Capflex (Fig. 6b, d). (c) It is well established that α-Syn amyloid aggregation (monitored using ThT) follows a sigmoidal growth kinetics with three distinct phases—the lag phase, the exponential phase and the saturation or plateau phase. To correlate the extent of reversibility of α-Syn droplets and the dilute phase concentration with the aggregation kinetics of α-Syn, the third set of samples were incubated with 10 μM of ThT and the aggregation kinetics was recorded with a plate reader in parallel to the Capflex measurements (Fig. 6g).

In Capflex, at 0 h, no peaks were observed in the Alexa channel with a stable baseline (at 100 μM) indicating that the sample was not phase separated (Fig. 6a). Our observation showed that after 4–8 h of incubation, small and sparse peaks started to appear in the Capflex traces, which we speculate could be due to possible nucleation events for α-Syn LLPS. The dilute phase concentrations also decreased slightly down to ~97–98 μM (Fig. 6a, middle panel). After 16 h, α-Syn phase separates into liquid droplets and the dilute phase concentration decreases to ~70 μM and remains almost unchanged until 24 h (Fig. 6a). The signal spikes completely disappeared when the phase-separated samples at 16, 20, and 24 h were subjected to dilution suggesting the liquid-like, reversible nature of the droplets (Fig. 6a, right panel). In the ThT channel, we observed no peaks for 0 and 10 h indicating no amyloid fibril formation. The 24 h samples showed few ThT positive peaks, which could be possibly due to the initiation of amyloid aggregation (Fig. 6b).

During time (30–48 h), the baseline fluorescence in the Alexa488 channel decreased significantly to ~40–50 μM with substantially more signal spikes—indicating more protein had partitioned into the dense phase (Fig. 6c). Interestingly, the signal spikes did not disappear even after diluting the sample (Fig. 6c, right panel). After 48 h, the baseline fluorescence reached ~2–5 μM, suggesting an irreversible liquid-to-solid transition and aggregation of α-Syn in the exponential phase. During this time (30–48 h), the baseline concentration increased substantially in the ThT channel with signal spikes—suggesting the presence of ThT positive, amyloid aggregates (Fig. 6d).

Microscopic observation of the droplets in bright field as well as in the fluorescent (Alexa488) channel showed presence of dynamic, liquid-like droplets at early time points (24 h) which gradually grows into larger assemblies with time (30–40 h) (Fig. 6e, f). Fluorescence microscopic observation using a ThT channel showed strong fluorescence enhancement of the dye in the droplets after 30 h, indicating the presence of amyloid aggregates (Fig. 6f, right panel). After 48 h, the droplets were found to be more sticky and a ThT positive, fibrillar morphology started to appear (Fig. 6f, right panel).

Strikingly, our plate reader aggregation assay using ThT showed that α-Syn LLPS solution (100 μM of α-Syn +20% (w/v) PEG6000 has a lag time of ~20–24 h. The exponential aggregation phase starts after ~20–24 h of incubation and continues until ~50 h. The system reaches saturation afterwards (Fig. 6g). Therefore, our data clearly shows that the onset of detectable α-Syn LLPS corresponds to the lag phase of aggregation. The dilute phase concentration remains stable for ~8 h after LLPS when the droplets grow by fusion and ripening. However, irreversible liquid-to-solid phase transition and amyloid aggregation happens primarily during the exponential phase (Fig. 6g, h) as confirmed by further decrease in the dilute phase concentration and ThT positive signal spikes after 30 h.

Interestingly, these experiments yielded virtually identical results when N-terminally acetylated (a physiologically relevant PTM[76–78]) fluorescently labeled (Alexa488) α-Syn A140C was used (Supplementary Fig. 12), confirming that the N-terminally acetylated protein partitions into the droplets similarly to the non-acetylated labeled protein during LLPS (Supplementary Fig. 12).

**Capflex throughput characterization**. The FIDA1 instrument can run two 96-well plates simultaneously. LLPS systems that require high temperature incubation (such as Ddx4n1 where the tray temperature is kept at 50 °C to prevent LLPS in the tray) might lead to sample evaporation during prolonged campaigns with the current setup of the instrument. Therefore, to demonstrate and calculate the throughput of Capflex, we have chosen α-Syn LLPS. We have analyzed LLPS behavior of α-Syn at a range of protein and PEG8000 concentrations at 37 °C. PEG8000 was chosen since the increased crowding effect (compared to PEG6000) induces instantaneous LLPS of α-Syn at high micromolar range[65]. We used 20, 50, 100, 150, and 200 μM α-Syn (in 20 mM PBS, pH 7.4) in presence of increasing (0, 10, and 20% (w/v) PEG8000 for an orthogonal screen of α-Syn LLPS (Supplementary Fig.13). The samples were prepared using the OpenTrons OT2 pipetting robot. The protein, buffer and PEG stocks were prepared manually. Subsequent experimental steps required no manual interventions. The sample preparation time by the OpenTrons OT2 robot was 36 min. Next, the samples were directly loaded in the FIDA1. For 15 samples used in our study, the FIDA1 run-time was measured as 180 min with a sample injection pressure of 2000 mbar. The measured throughput was calculated as $r = 15$ samples/3.6 h = ~4 samples/h = 96 samples/day including sample preparation time with no human supervision. Each sample vial contained 40 μl of sample for two iterations/sample. Therefore, the total sample volume to run this orthogonal LLPS screen was only 0.6 ml.

Our results showed instantaneous (0 h) α-Syn LLPS at 200 μM concentrations in presence of 10 and 20% (w/v) PEG8000 (Supplementary Fig. 13) with a partitioning of ~30 μM dilute phase equivalent α-Syn in the droplet phase. We have demonstrated this by plotting the [total protein concentration − dilute phase concentration] (Supplementary Fig. 13). The non-phase-separated samples showed ±7 μM difference in the total protein concentration, which was within the range of error. Together, our results demonstrates an individual campaign at HTP with minimal sample consumption and human supervision. Additionally, from this experiment, we note that the technique also provides non-binary phase diagrams by measuring the dilute phase concentration.

## Discussion

As the evidence for the important role of LLPS in functional and deleterious biological process is mounting[3], there is an increasing

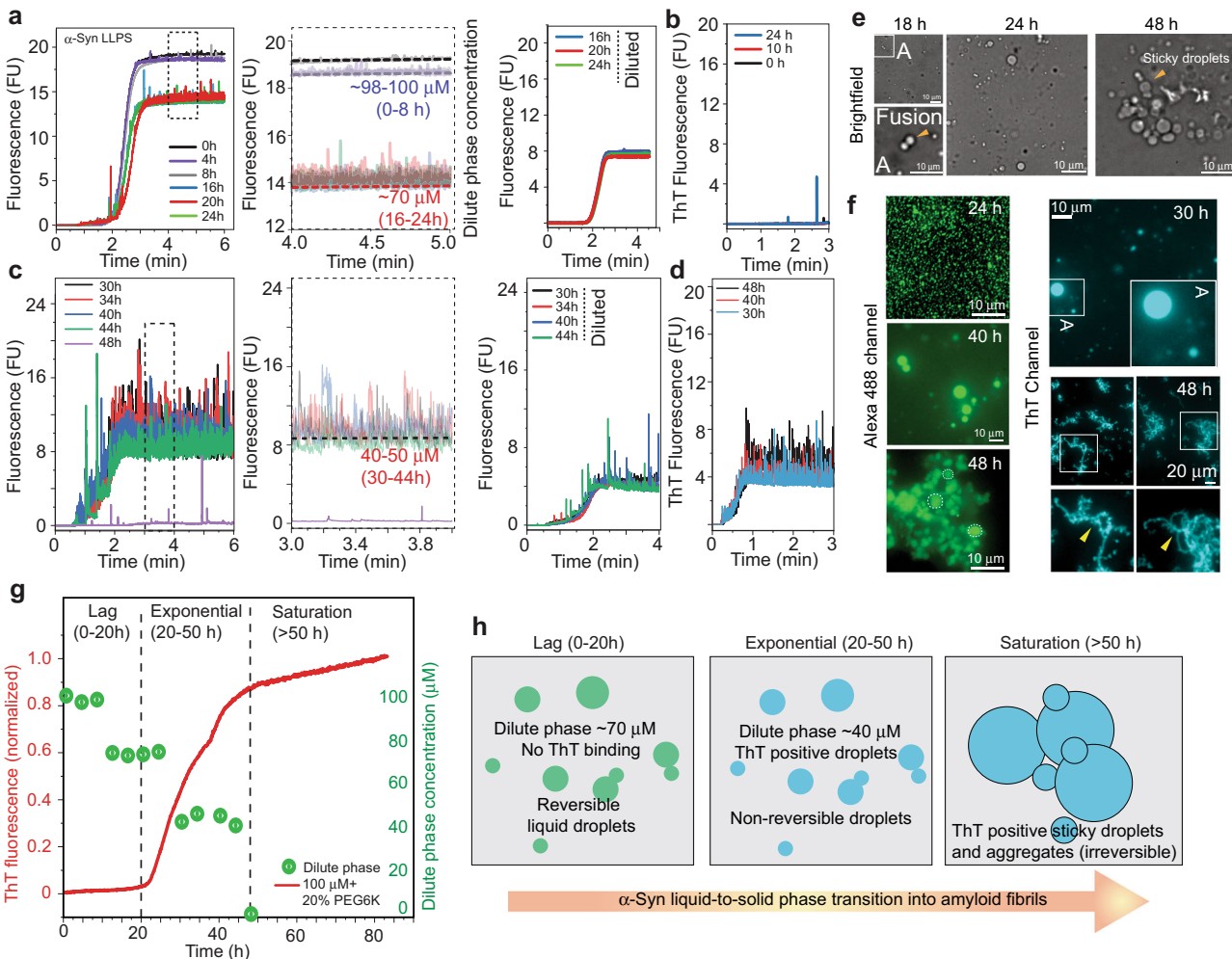

**Fig. 6 Simultaneous monitoring of α-Syn LLPS and subsequent amyloid aggregation using Capflex. a** (Left panel) Capflex measurements of α-Syn LLPS samples spiked with 10 nM Alexa488 labeled α-Syn until 24 h is shown. The appearance of peaks with time indicates LLPS of α-Syn. (Middle panel) A magnified view of the fluorescence readouts from Capflex showing the evolution of dilute phase concentration during α-Syn LLPS. (Right panel) Until 24 h, the reversibility/dissolution of α-Syn LLPS upon dilution with buffer is confirmed by the disappearance of signal spikes in Capflex. The data is obtained from the same samples as the one in the left and middle panel. **b** Capflex readouts in ThT channel for 0, 10, and 24 h samples. **c** (Left panel) Capflex measurements of α-Syn LLPS samples (Alexa488 channel) from 30 to 48 h. (Middle panel) A magnified view of the fluorescence readouts from Capflex showing the dilute phase concentrations during 30–48 h incubation. (Right panel) The signal spikes do not disappear upon dilution with buffer indicating the irreversible nature of the system after 30 h. **d** ThT signal spikes are observed for the samples after 30 h suggesting the presence of amyloid aggregates. Representative Capflex traces are reported. $n = 2$ independent experiments (**a–d**). **e** Bright field microscopy images of liquid droplets after 18 h. A fusion event is magnified and indicated with a yellow arrowhead. $n = 2$ independent experiments. **f** (Left panel) fluorescence microscopy images of Alexa488 labeled α-Syn droplets at 24, 40, and 48 h. (Right panel) fluorescence microscopy images of α-Syn droplets spiked with 10 μM ThT at 30 and 48 h. Amyloid fibril like morphology is observed only after 48 h. **g** Normalized ThT fluorescence showing amyloid aggregation of 100 μM α-Syn (in 10 mM PBS, 200 mM NaCl, pH 7.4) in the presence of 20% (w/v) PEG6000 (red) is overlaid with the dilute phase concentration obtained from Capflex measurements (green). The lag, exponential and saturation phases are marked with black dashed lines. $n = 2$ independent experiments. **h** Schematic depicting the aberrant phase transition of α-Syn LLPS from a reversible state to a non-reversible state. Source data are provided as a Source Data file.

need for quantitative and widely accessible experimental methodology that allows to characterize the key parameters of LLPS. Various experimental methods exist to detect droplet formation (such as microscopy, turbidity[52]), quantify droplet size distribution (microscopy[52]) and dilute phase concentration (UV-Vis absorbance spectroscopy[52]) and to draw complete phase diagrams (e.g. microfluids water-in-oil emulsion droplet systems[53–55,79]). However, most of these methods suffer from one or more inconveniences, such as low throughput (microscopy), large sample requirements (direct measurement of dilute phase concentration after centrifugation) or a high technical skill barrier (microfluidics). Here we present Capflex, a method that allows all of the above mentioned properties crucial to LLPS to be

quantitatively measured (Fig. 1). Conventional approaches also often require multiple instances of human intervention and supervision and therefore can be person-time consuming. Moreover, the determination of droplet size distributions requires extensive microscopic measurements followed by laborious analysis of the droplet sizes. On the other hand, Capflex can independently be used to measure all the said parameters with little human intervention (person-time) and provides a significant information content with minimal sample consumption. Capflex is simple and based on a commercially available turnkey instrument that has been designed and developed for biomolecular interaction analysis in solution by Taylor dispersion analysis[60,69]. This functionality allows to also quantify the interactions between

a protein undergoing LLPS and a modulating compound, such as DNA if these interactions lead to a measurable change in size of the complex. We have exploited this feature here in the context of LLPS of Ddx4n1 and RP$_3$ (Figs. 2–4), where we have measured the binding affinity to DNA for both systems. Nucleic acids are one of the most important physiological control parameters for LLPS[33], and therefore the ability to quantify both the effect of nucleic acids on LLPS as well as the molecular binding affinity in the dilute (non LLPS) regime is of great benefit and interest. A deleterious effect on droplet stability is observed at concentrations approximately one order of magnitude below the precisely determined affinity (by FIDA1) in the dilute regime between DNA and protein. Therefore, if one wanted to estimate an affinity of the droplet state for the DNA, it is likely to be approximately one order of magnitude higher than the dilute phase affinity. However, the stoichiometry of interactions with the droplet phase is probably very different, because of the possibility of the DNA to simultaneously interact with several protein molecules at the same time (Fig. 2). This difference reflects the fact that a client molecules, e.g. DNA can not only act as *glue* for LLPS formation (as for RP$_3$), but also as efficient disruptors of LLPS (Figs. 2 and 4).

Using Capflex, we were able to study the influence of additives, such as Ca$^{+2}$ and Mg$^{+2}$ ions, and the molecular crowding agent PEG on the driving force for droplet formation by Ddx4n1 in a straightforward manner, using only 10 µl of sample for each data point (Figs. 1 and 3). Furthermore, we were able to resolve differences in droplet formation kinetics as a function of PEG concentration and hence driving force (Fig. 3), with an instrument dead time of ~15 s, as well as the changes in droplet size distribution over time (Fig. 3). LLPS is an inherently out-of-equilibrium phenomenon; assuming the droplet state corresponds to the free energy minimum, equilibrium corresponds to a single large drop of dense phase being in contact with the dilute phase[4]. It is therefore of interest to be able to monitor the relaxation of the system to this equilibrium state through droplet fusion and Ostwald ripening.

We also studied the LLPS and subsequent amyloid fibril formation by the protein α-Syn (Figs. 5 and 6), where we exploit the fact that the standard fluorescence channel of the instrument is able to detect also the emission by the dye ThT when bound to amyloid fibrils. Differently to Ddx4n1 and RP$_3$, LLPS of α-Syn is exceedingly slow[65] under the conditions we have emploied. Reliable droplet formation is only observed after 16–18 h. By that point, droplets are still clearly liquid and merge and dissolve easily. Only negligible ThT emission is detected at this point, and the dilute phase concentration is ~70 µM (Fig. 6). However, another 24 h later, we find that the dilute phase concentration has decreased by more than one order of magnitude. This decrease in background concentration is associated with a very strong increase in ThT fluorescence, both in the form of a constant background, as well as individual large spikes (Fig. 6). This dual signal probably reflects a background of individual fibrils too small to be resolved by Capflex, in addition to some larger clusters of fibril. The possibility to follow both the decrease in background concentration and the increase in ThT fluorescence allows to correlate these two variables. It is highly likely that the conversion of the reversible liquid droplets into amyloid fibrils is causing the significant decrease in background concentration, that we quantify here. The objective of the present work is to demonstrate the insight that can be gained from quantitative measurements of LLPS and to present a powerful and easy-to-use platform to achieve such measurements, Capflex.

## Methods

**Reagents and chemicals.** pET30M-2 plasmids encoding His_tag-GST-TEV_site-Ddx4n1 or His_tag-GST-TEV_site-Ddx4n1-YFP (see sequence details below) were kind gifts from Dr. Tim Nott (Oxford University). Ampicillin resistant TEV plasmid encoding MBP–TEV_site–His-tag-S-TEV was a kind gift from Charlotte O'Shea (University of Copenhagen). All constructs where sequenced by GATC (Eurofins, Germany) in order to ensure no point mutations had occurred. Tris, TCEP, NaCl, CaCl$_2$, PEG3000, PEG6000, Imidazole, LB broth, benzonase, ampicillin, kanamycin and reduced glutathione were all purchased from VWR (Denmark). The ssDNA as previously described to be partitioned by Ddx4n1 droplets by Nott et al.[35] with the sequence 5′-TTT TTC CTA GAG AGT AGA GCC TGC TTC GTG G-3′ as well as the 5′Alexa-488 labeled version was synthesized, HPLC purified and lyophilized by TAG Copenhagen (Denmark). The RP$_3$ peptide was synthesized and HPLC purified by Bachem (Switzerland) and delivered as TFA salt after MALDI-MS quality control. Fluoresbrite YG Microspheres of calibration grade were purchased from Polysciences Europe GmbH (Germany). All other relevant salts, buffer components and materials were purchased from Sigma (USA) & VWR (Denmark) and dissolved in RNAse and nuclease free milliQ water unless otherwise specified.

**Protein expression and purification**

*Ddx4-n1 and Ddx4n1-YFP.* Overnight cultures of *E. coli* BL21 (DE3) carrying the plasmid encoding the MBP–TEV_site–His-tag-S-TEV were inoculated into 1 L LB-Amp media in a 3 L flask and grown at 37 °C at 160 rpm shaking. When OD$_{600}$ = 0.8 was reached the expression of the construct was induced by addition of IPTG to a final concentration of 1 mM. The induced cells were incubated overnight for 16 h at 20 °C at 160 rpm shaking. The cells were harvested by centrifugation at 4 °C, 7000 × g, 20 min and stored at −20 °C until use. The expressed MBP-TEV construct undergoes immediate auto cleavage upon expression. Cells were resuspended in 50 ml 20 mM Tris pH 7.8, 200 mM NaCl, 1% Glycerol at 4 °C, and lysed by sonication for 12 × 30 son ice. The lysate was centrifuged for 20 min at 20,000 × g, 4 °C, and the supernatant was loaded onto 2 ml Ni-NTA resin equilibrated with 20 mM Tris pH 7.5, 200 mM NaCl at 4 °C. The resin and supernatant were incubated at 4 °C for 1 h under gentle mixing fast enough for the resin not to sediment. The resin was washed with 20 mM Tris pH 7.5, 100 mM NaCl, 20 mM imidazole and the TEV protease was eluted with 8 ml 20 mM Tris pH 7.5, 100 mM NaCl, 250 mM imidazole. The eluted TEV protease was dialyzed against 2 L 50 mM Tris pH 7.5, with 100 mM NaCl, 10 mM β-mercaptoethanol, 1 mM EDTA at 4 °C using 6 kDa cut-off membrane. Glycerol was added to the protein to a final concentration of 50% and it was stored at −80 °C until use.

Expression of Ddx4n1 constructs was carried out by transforming the pET30M-2 carrying the Ddx4n1 constructs into *E. coli* BL21 (DE3). Overnight cultures of these where inoculated into 4 L of LB-KAN media in 4 3 L flasks and incubated at 37 °C at 160 RPM until an OD$_{600}$ between 0.6 and 0.8 was reached. The expression was then induced by addition of IPTG to a final concentration of 0.5 mM. The temperature was lowered to 20 °C and the expression took place overnight for 16 h. The cells were harvested by centrifugation at 7000 × g, 4 °C, for 20 min and stored at −20 °C until use. Cells from 4 L culture were resuspended in 120 ml 50 mM Tris pH 8.0, 500 mM NaCl, 5 mM DTT on ice and 16 µL benzonase was added to remove DNA. The cells were lysed by sonication for 12 × 30 s on ice, centrifuged (20,000 × g, 4 °C, for 20 min) and the supernatant was loaded onto 20 ml Pierce Superflow GST-agarose equilibrated with 50 mM Tris, 500 mM NaCl, pH 8.0. The resin was incubated under gentle mixing for 30 min at 4 °C. The bound protein was washed with 50 mM Tris, 500 mM NaCl, pH 8.0 and eluted with 60 ml 50 mM Tris, 500 mM NaCl, pH 8.0 containing 10 mM reduced glutathione, all at 4 °C. The eluted protein was stored in the fridge overnight. EDTA and DTT were added to the eluted protein to a final concentration of 1 and 2 mM respectively. 3 ml of the purified TEV protease was added and the sample was incubated at ambient temperature for 3 h under gentle mixing. The cleaved Ddx4n1 constructs were purified by Reverse IMAC by loading the sample onto 10 ml Ni-NTA resin equilibrated with 25 mM Tris, 250 mM NaCl, 20 mM Imidazole. The Ni-NTA resin binds both the TEV protease and the cleaved GST tag. The resin was incubated under gentle mixing for 30 min and the flow through was collected, concentrated to 1 ml using 10 kDa cut-off spin filters. 1 ml of the concentrated protein was loaded onto a Superdex 75 16/60 column (GE healthcare) equilibrated with 20 mM Tris pH 8.0, 300 mM NaCl, 5 mM TCEP and the protein was eluted at a flow rate of 1 ml/min at 7 °C. Protein purity was assessed by SDS-PAGE and MALDI-TOF operating in linear positive mode (Ultraflex II, Bruker Daltonics). Protein concentrations were assessed by measurement of the absorption at 280 nm and the concentrations calculated using theoretical extinction coefficients predicted by ProtParam[80] (Expasy, Switzerland). The verification of the absence of DNA and of the reproducibility of cloud points between batches was performed using a multichannel spectrophotometer/fluorimeter with thermal control (ProbeDrum, ProbationLabs, Sweden). The purified Ddx4n1 and Ddx4n1YFP proteins were concentrated to 500 and 320 µM respectively, aliquoted and stored at −80 °C until use. Total protein yields from 4 L culture were up to 11 mg.

*Wild-type (WT) α-Syn expression and purification.* Overnight cultures of *E. coli* BL21 (DE3) carrying the pT7-7 plasmid encoding the WT human α-Syn gene were inoculated into 1 L LB-Amp media in a 3 L flask and grown at 37 °C at 180 rpm shaking. When OD$_{600}$ = 0.8 was reached the expression of the construct was induced by addition of IPTG to a final concentration of 1 mM. The induced cells were incubated for 4 h at 37 °C at 180 rpm shaking. The cells were harvested by

centrifugation at 4 °C, 7000 × g, 20 min and stored at −20 °C until use. Bacterial pellet corresponding to 1 L culture was resuspended in 20 ml of 10 mM Tris-HCl, 1 mM EDTA, pH 8.0 with 1 mM PMSF. The suspension is sonicated with a probe sonicator for 2 min at 10 s of sonication, 30 s pause, 12 rounds, 40% amplitude. 1 μL Benzonase was added to the cell lysate and centrifuged at 20,000 × g, 30 min at 4 °C to remove the DNA. The supernatant is collected, and the solution is boiled for 20 min. The solution was subsequently centrifuged at 20,000 × g for 20 min at 4 °C in order to precipitate the heat-sensitive proteins, α-Syn remaining in the supernatant. Next, 4 ml saturated $(NH_4)_2SO_4$ was added per 1 ml supernatant to salt out α-Syn. The solution was stirred at 4 °C for 15 min and centrifuged at 20,000 × g for 20 min, at 4 °C to pellet down the protein. The pellet is dissolved in 7 ml of 25 mM Tris-HCl pH7.7 and 7 μl DTT is added to a final concertation of 1 mM. Next, the protein solution was dialyzed against the same buffer for 16–18 h at 4 °C with a change of the buffer after 12 h of dialysis. The dialyzed protein solution was then subjected to anion exchange column (AEC) (HiTrap Q Hp 5 ml, GE healthcare) followed by size exclusion chromatography (SEC) (HiLoad 16/600 Superdex 200 pg. column) and eluted in 10 mM of sodium phosphate buffer (pH 7.4). Protein concentrations were assessed by measurement of the absorption at 280 nm (with the help of a spectrophotometer—ProbeDrum, ProbationLabs, Lund, Sweden) and the concentrations calculated using theoretical molar extinction coefficients predicted by ProtParam[80] (Expasy, Switzerland).

*Non-acetylated and acetylated α-Syn-A140C expression and purification.* For non-acetylated α-Syn-A140C, BL21 DE3 pLysS competent *E. coli* was transformed with pT7-7 plasmid containing the gene encoding for α-Syn-A140C. The cells were selected on LA plates using ampicillin (100 μg/mL) marker. The cells were subsequently cultured, and protein expression was induced using 1 mM of Isopropyl β-d-1-thiogalactopyranoside (IPTG) for 4 h at 37 °C. The cells were harvested following protein expression and cell pellets were re-suspended in osmotic shock buffer (10 mM Tris-HCl, 1 mM EDTA, pH 8.0, 1 mM PMSF with 1 mM DTT to prevent intermolecular disulfide linkage of the A140C α-Syn) and homogenized for 45 min at 4 °C. The homogenate was centrifuged for 20 min at 20,000 × g at 4 °C. The supernatant was boiled for 15 min in a water bath and cooled down on ice. The pellet from the centrifugation step, was re-suspended in ice cold MQ water with the addition of 2.5 mM MgCl₂. The solution was subsequently homogenized for 30 min and the homogenate was centrifuged at 20,000 × g for 20 min at 4 °C. This was followed by boiling and re-cooling as previously described. The samples were pooled together and centrifuged at 20,000 × g for 20 min at 4 °C. The final supernatant was collected and dialyzed for 2 h (with one buffer exchange after 1 h) against 25 mM Tris, pH 7.5, followed by addition of Streptomycin sulfate (10 mg/mL) for 15 min in 4 °C to precipitate DNA. Finally, α-Syn-A140C was precipitated with saturated $(NH_4)_2SO_4$ and dialyzed as mentioned in the previous section. The protein was purified using filtration (using a 0.44 μm membrane) followed by AEC and SEC. IEC was done loading the samples onto a pre-equilibrated HiTrap Q FF column using an external Bio-Rad econo pump at RT. After loading the column, α-Syn-A140C was eluted with a salt gradient (25 mM Tris, pH 7.5, 1 M NaCl) on an Äkta purifier and all samples were verified by SDS-PAGE. Only the fractions containing α-Syn-A140C monomer were collected and was further purified using a HiLoad 26/600 Superdex 75 pg column equilibrated in 20 mM sodium phosphate buffer, pH 6.5. All buffers for purification of α-Syn-A140C contained 1 mM DTT and 1 mM EDTA to prevent intermolecular cysteine-disulfide linkages.

For the N-acetylated α-Syn-A140C, BL21 DE3 pLysS competent *E. coli* was co-transformed with pT7-7 plasmid containing the gene encoding for α-Syn-A140C along with pNatB plasmid (pACYCduet-naa20-naa25) for N-acetyl transferase enzyme, both under IPTG inducible lac promoters. The selection was carried out on LA plates using ampicillin (for α-Syn-A140C containing plasmid) and chloramphenicol (for pNatB plasmid) selection markers. The rest of the purification protocol for the N-acetylated α-Syn-A140C was identical to that of non-acetylated α-Syn-A140C.

*Alexa488 labeling of α-Syn-A140C.* The protein sample was first concentrated to 13.655 mg/ml for sufficient labeling. For separation of DTT and EDTA from α-Syn-A140C, the sample was injected into a Superdex 200 increase, 10/300 GL column connected to an Äkta purifier. The fraction containing α-Syn-A140C was collected, pooled (5.813 mg/ml) and used for downstream labeling. AlexaFluor-488 C-5 maleimide (Invitrogen) was dissolve in DMSO (1 mg in 200 μl) yielding a 6.9 mM dye concentration. The conjugation reaction was done by incubating 500 μl of 358 μM α-Syn-A140C with 200 μl AlexaFluor-488 C-5 maleimide (10-fold excess of dye) for 1 h at room temperature. The reaction solution was added to the Superdex 200 increase, 10/300 GL column, where the free dye and the conjugated Alexa488-α-Syn-A140C were separated, and the concentration was calculated according to the absorbance at 275 nm (α-Syn-A140C) and 488 nm (Alexa488) using a molar absorption coefficient (ε) = 5960 M⁻¹ cm⁻¹ for α-Syn-A140C. The labeling protocol for N-acetylated α-Syn-A140C was identical to the non-acetylated α-Syn-A140C.

## LLPS analysis with FIDA 1 by capillary flow experiments (Capflex)

*General description of the FIDA1 instrument.* The FIDA 1 instrument consists of an autosampler holding two 96-well plates whose temperature can be individually controlled. The autosampler loads the sample directly into the 1 m long capillary that is housed in a separate thermostatted chamber. Three modes of operation are

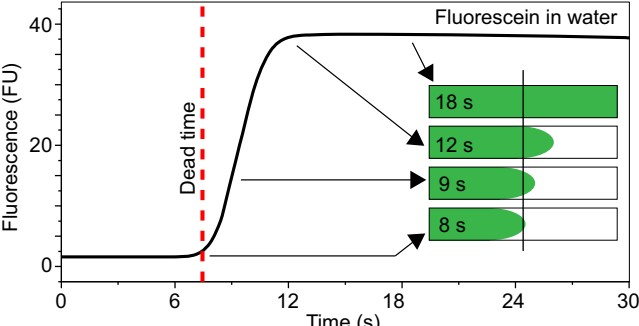

**Fig. 7 Origin of sigmoidal traces in Capflex.** 1 nM Fluorescein in water is measured in Capflex at 3500 mbar injection pressure. The dead time of the instrument is the time it takes for the solution front to reach the detector window (black line of the scheme in the inset). The dead time of ~8 s is marked with a dashed red line in the actual data. The schematic depicts the progress of the solution front with time resulting in a sigmoidal increase of fluorescence intensity, which reaches a stable baseline fluorescence after 12 s when the solution front has completely crossed the detector window. Source data are provided as a Source Data file.

used in this study, (1) standard Capflex (i.e. slow continuous injection of sample), (2) rapid filling of the capillary followed by slow continued injection of the sample for the study of LLPS kinetics and (3) standard FIDA for affinity determination. Notably, the dead time of the instrument (time taken for the solution front to reach the detector window) depends on the sample viscosity and sample injection pressure and therefore, is not a constant quantity. The extent of phase separation, which may lead to an increase in viscosity, also results in an increased dead time for Capflex for a given injection pressure. We demonstrated the dead time of our instrument for a sample having similar viscosity to water (1 nM Fluorescein in water). The calculated dead time was ~8 s at 3500 mbar of pressure (Fig. 7). The origin of the sigmoidal shape of Capflex traces is explained schematically in Fig. 7. Briefly, the sigmoidal pattern arises from the shape of the solution front passing the detector. In the parabolic Poiseuille flow profile of the FIDA1 capillary, the fluid in the center of the capillary flows faster than the fluid close to the walls. Due to this flow profile, the central part of the solution reaches the detector window first—leading to the sigmoidal increase in fluorescence signal (Fig. 7).

## Standard Capflex analysis

To perform this, the capillary is equilibrated with sample buffer and the sample is then loaded into the capillary. The dead time here can be controlled by the applied pressure, but should preferably be long enough that a pseudo-equilibrium for LLPS has been achieved, i.e. that the dilute phase concentration has reached its equilibrium value and can be read from the fluorescence baseline. If the fluorescence signal stems from a small concentration of added labeled protein, the dilute phase concentration can be defined under the assumption that the labeled and unlabeled protein molecules partition into the droplets in the same manner. In future-versions of the instrument, it will also be possible to detect intrinsic fluorescence and hence unlabeled samples can be used.

## Kinetic analysis

The capillary is loaded at the highest possible pressure (3500 mbar) resulting in a dead time of ~8 s for a sample with viscosity similar to that of water. The measurement is then started, the pressure lowered and the kinetic development of the sample is followed.

## Affinity determination[60]

The capillary is filled with a solution of interaction partner of known concentration. A small injection of 40 nl of the labeled biomolecule + interaction partner is then carried out and it is pushed through the capillary by the solution of interaction partner. The laminar flow in the capillary causes deformation of the plug, which is measured by a fluorescence detector. The degree of this deformation depends on the diffusivity of the labeled biomolecule. Hence, diffusivity is directly measured which can be converted to hydrodynamic radius through the Stokes–Einstein relation. The change in hydrodynamic radius can then be fitted to the binding models described in the following section.

**Ddx4n1 LLPS:** For Ddx4n1 LLPS experiments, the sample chamber containing the glass vials with 30 μL sample was kept at a temperature above the cloud point of the respective set of solution conditions. PEG3000 was added to 50 μM Ddx4n1 in 20 mM Tris pH 8.0, 100 mM NaCl, 5 mM TCEP to a final concentration of 0–7%. The influence of CaCl₂ and MgCl₂ was investigated at a concentration of 100 μM Ddx4n1 and a bivalent salt concentration between 0 and 10 mM. Influence of protein concentration was investigated at a Ddx4n1 concentration between 100 and 163 μM. The influence of ssDNA was investigated at a concentration of 128 μM Ddx4n1 and an ssDNA concentration between 0

**Table 1 Experimental parameters for Capflex analysis of Ddx4n1 on the FIDA 1 instrument.**

| Tray | Vial | Pressure (mbar) | Time (s) | Outlet | Measure | Comment |
|---|---|---|---|---|---|---|
| 2 | 1 | 3500 | 45 | Variable | No | 1 M NaOH wash |
| 2 | 2 | 3500 | 60 | Variable | No | milliQ wash |
| 1 | 48 | 3500 | 60 | Variable | No | 1% Tween coat |
| 1 | Indicator | 3500 | 40 | Variable | No | Buffer wash |
| 1 | Analyte | 200 | 540 | Variable | Yes | Sample application |

Trays 1 and 2 were maintained at 50 °C and the capillary chamber was maintained at 20 °C.

**Table 2 Experimental parameters for Capflex analysis of RP₃ on the FIDA 1 instrument.**

| Tray | Vial | Pressure (mbar) | Time (s) | Outlet | Measure | Comment |
|---|---|---|---|---|---|---|
| 2 | 1 | 3500 | 40 | Variable | No | 300 mM NaCl wash |
| 2 | 2 | 3500 | 40 | Variable | No | milliQ wash |
| 1 | Indicator | 3500 | 40 | Variable | No | Buffer wash |
| 1 | Analyte | 400 | 300 | Variable | Yes | Sample application |

Trays 1 and 2 were maintained at 20 °C and the capillary chamber was also maintained at 20 °C.

**Table 3 Experimental parameters for Capflex analysis of α-Syn on the FIDA 1 instrument.**

| Tray | Vial | Pressure (mbar) | Time (s) | Outlet | Measure | Comment |
|---|---|---|---|---|---|---|
| 2 | 1 | 3500 | 45 | Variable | No | 1 M NaOH wash |
| 2 | 2 | 3500 | 60 | Variable | No | milliQ wash |
| 1 | Indicator | 3500 | 40 | Variable | No | Buffer wash |
| 1 | Analyte | 1200 or 2000 | 600 | Variable | Yes | Sample application |

Trays 1 and 2 were maintained at 37 °C and the capillary chamber was also maintained at 37 °C. Since this system consisted of 20% PEG6000, due to high viscosity, the sample application was performed with a relatively higher pressure (1200 or 2000 mbar).

**Table 4 Experimental parameters for HTP screen of α-Syn LLPS on the FIDA 1 instrument.**

| Tray | Vial | Pressure (mbar) | Time (s) | Outlet | Measure | Comment |
|---|---|---|---|---|---|---|
| 2 | 1 | 3500 | 45 | Variable | No | 1 M NaOH wash |
| 2 | 2 | 3500 | 60 | Variable | No | milliQ wash |
| 1 | Indicator | 3500 | 40 | Variable | No | Buffer wash |
| 1 | Analyte (LLPS samples) | 1200 or 2000 | 360 | Variable | Yes | Sample application |

Trays 1 and 2 were maintained at 37 °C and the capillary chamber was also maintained at 37 °C. Since this system consisted of 20% PEG8000, due to high viscosity, the sample application was performed with a relatively higher pressure (1200 or 2000 mbar). The table represents the parameters set for one single run out of 15 total runs.

and 30 µM, with the addition of the fluorescent Ddx4n1YFP. Ddx4n1-YFP was added to the 500 µM Ddx4n1 stock solution to a final concentration of 200 Nm (0.5–20 µl). When the Alexa488 labeled DNA was used as indicator it was at a Ddx4n1 concentration of 112 µM, 0–26 µM ssDNA and 0.5% labeled DNA. Fluoresbrite spheres were diluted 4000 times for the 6 µm spheres and 400,000 times for the 1 µm spheres in 20 mM Tris pH 8.0, 100 mM NaCl prior to Capflex analysis. For all samples, a capillary of inner diameter of 75 µm and a length of 1 m was used.

The following set of experimental parameters was used for all samples (Table 1):

**RP₃ LLPS**: For the RP₃ peptide the cloud point is above the maximum temperature reachable by the FIDA 1 instrument. Hence, the samples were mixed immediately prior to injection. An RP₃ concentration of 184 µM and a DNA concentration of 3.3–40 µM in 10 mM Tris pH 8.0, 50 mM NaCl was used for the analysis. All experiments were carried out with 0.17% labeled ssDNA. The capillary was coated using HS-coating (FIDAbio, Denmark) in order to avoid peptide interactions with the capillary surface.

The following set of experimental parameters was used for all samples (Table 2):

**α-Syn LLPS**: 100 µM of α-Syn (in 10 mM sodium phosphate buffer (Na₂HPO₄•2H₂O and NaH₂PO₄•H₂O), pH 7.4, 200 mM NaCl) was subjected to LLPS in presence of 20% (w/v) PEG6000[64]. The samples were incubated at 37 °C to monitor the formation and maturation of the droplets. For Capflex experiments, 5 nM of Alexa488 labeled α-Syn A140C and N-acetylated α-Syn A140C were used in the reaction mixtures as fluorescent indicators. For ThT measurements, 5 µM ThT was added to the LLPS samples. Each tube was incubated at standstill conditions and sealed with parafilm to avoid possible evaporation during long incubation times. 50 µl of sample was aliquoted for each Capflex measurements. For HTP screening of α-Syn LLPS (Supplementary Fig. 13), samples of

20, 50, 100, 150, and 200 µM α-Syn was prepared in 20 mM PBS, pH 7.4 (commercially available tablets, Sigma, USA) in the presence of increasing (0, 10, and 20% (w/v) PEG8000 concentration. The solutions were spiked with 50 nM (0.025%) Alexa488-maleimide labeled A140C α-Syn.

The following set of experimental parameters was used for all samples (Table 3):

To note, the experimental parameters for the HTP screening of α-Syn LLPS is provided in Table 4.

**Automated peak and baseline analysis**. For all three LLPS systems used in our study, the baseline and peak height determination were carried out with a python script running in Jupyter 6.1.4 (available here: https://doi.org/10.11583/DTU.14223116) with a peak threshold of 0.2 from the baseline. The start of the signal baseline was defined as the top line of a sigmoid fit (>99%) to the time dependent fluorescence trace from the capillary. Prior to fitting the model, this line was smoothed by removing signal larger than +10% of the expected signal predicted by the standard curve and then applying a median filter with a windows size of 31 data points. The baseline fluorescence was then calculated as the lower 5% quantile of this line. The baseline fluorescence intensity was converted into a protein, peptide or DNA concentration by running standard samples of known concentration that had not undergone LLPS.

**Affinity and complex size determination using FIDA 1**. Affinity of Ddx4n1-YFP for ssDNA was determined by standard FIDA analysis[60]. 400 nM Ddx4n1-YFP in

**Table 5 Method for FIDA analyses for affinity determination of Ddx4n1 and ssDNA.**

| Tray | Vial | Pressure (mbar) | Time (s) | Outlet | Measure | Comment |
|---|---|---|---|---|---|---|
| 2 | 1 | 3500 | 45 | Variable | No | 1 M NaOH wash |
| 2 | 2 | 3500 | 60 | Variable | No | milliQ wash |
| 1 | 48 | 3500 | 60 | Variable | No | 1% Tween coat |
| 1 | 46 | 3500 | 40 | Variable | No | Buffer wash |
| 1 | Analyte | 3500 | 20 | Variable | No | Analyte capillary fill |
| 1 | Indicator | 50 | 10 | Variable | No | Indicator-plug inject |
| 1 | Analyte | 400 | 180 | Variable | Yes | Mobilize and measure |

Tray 1 and 2 was maintained at 20 °C and the capillary chamber was also maintained at 20 °C for all our measurements.

**Table 6 Method for FIDA analyses for affinity determination of RP₃ and ssDNA.**

| Tray | Vial | Pressure (mbar) | Time (s) | Outlet | Measure | Comment |
|---|---|---|---|---|---|---|
| 2 | 1 | 3500 | 45 | Variable | No | 1 M NaOH wash |
| 2 | 2 | 3500 | 75 | Variable | No | Buffer wash |
| 1 | Analyte | 3500 | 20 | Variable | No | Analyte capillary fill |
| 1 | Indicator | 50 | 10 | Variable | No | Indicator-plug inject |
| 1 | Analyte | 400 | 180 | Variable | Yes | Mobilize and measure |

Tray 1 and 2 was maintained at 20 °C and the capillary chamber was also maintained at 20 °C for all our measurements.

**Table 7 Experimental parameters for Capflex analysis of kinetics of droplet formation on the FIDA 1 instrument.**

| Tray | Vial | Pressure (mbar) | Time (s) | Outlet | Measure | Comment |
|---|---|---|---|---|---|---|
| 2 | 1 | 3500 | 45 | Variable | No | 1 M NaOH wash |
| 2 | 2 | 3500 | 60 | Variable | No | milliQ wash |
| 1 | 48 | 3500 | 60 | Variable | No | 1% Tween coat |
| 1 | 46 | 3500 | 40 | Variable | No | Buffer wash |
| 1 | Analyte | 3500 | 20 | Variable | No | Analyte capillary fill |
| 1 | Analyte | 300 | 240 | Variable | Yes | Kinetics measure |

Trays 1 and 2 were maintained at 55 °C and the capillary chamber was maintained at 20 °C.

20 mM Tris pH 8.0, 100 mM NaCl, 5 mM TCEP was used as indicator, more than 2 orders of magnitude bellow the lower critical concentration of LLPS. This binding affinity measurement was performed with a constant, low protein concentration that was titrated with increasing concentrations of ssDNA, because the inverse experiment suffers from complications when high protein concentrations are reached that undergo LLPS. ssDNA concentration of 0–284 μM was added to both indicator and analyte. All samples were measured in triplicates. The following set of instrumental parameters was used (Table 5):

Apparent hydrodynamic radius ($R_h$) was determined by analyzing the resulting Taylor grams[69,70] with the FIDA software suite (FIDAbio, Denmark). Notably, 1 to 1 (Eq. 1) or 1 to 2 (Eq. 2) binding model was fitted to the resulting data.

$$R_h = \frac{\left(1 + \frac{1}{K_d} * [analyte]\right)}{\left(R_{h_{unbound}}{}^{-1} - R_{h_{bound}}{}^{-1}\right) + \left(1 + \frac{1}{K_d} * [analyte]\right) * R_{h_{bound}}{}^{-1}} \tag{1}$$

$$R_h = \frac{\left(1 + \left(\frac{1}{K_d}\right)^2 * [analyte]^2\right)}{\left(R_{h_{unbound}}{}^{-1} - R_{h_{bound}}{}^{-1}\right) + \left(1 + \left(\frac{1}{K_d}\right)^2 * [analyte]^2\right) * R_{h_{bound}}{}^{-1}} \tag{2}$$

Were $R_h$ is the apparent hydrodynamic radius of the sample, $K_d$ the dissociation constant, $R_{h\_unbound}$ and $R_{h\_bound}$ were the apparent hydrodynamic radii of the free and complexed form of the indicator, respectively[60,81].

For RP₃, Alexa488 labeled ssDNA (L-DNA) was used as indicator at a concentration of 83 nM and RP₃ peptide was added at a concentration of 0–358 μM in 10 mM Tris-HCl and 50 mM NaCl. A 1 to 2 (Eq. 2) binding model was fitted to the resulting data (Eq. 2)

$$R_h = \frac{\left(1 + \left(\frac{1}{K_d}\right)^2 * [analyte]^2\right)}{\left(R_{h_{unbound}}{}^{-1} - R_{h_{bound}}{}^{-1}\right) + \left(1 + \left(\frac{1}{K_d}\right)^2 * [analyte]^2\right) * R_{h_{bound}}{}^{-1}}$$

Both models were fitted using Origin Pro 2019 software suite (Origin labs, USA). The model with two binding sites assumes full cooperativity (Hill exponent 2) and does not include the molecular species with only one ligand bound. The reason for this simplification is that the experimental data only provides well-defined sizes for the species with 0 and all (2) ligands bound. Inclusion of a species with 1 ligand bound and a separate hydrodynamic radius would lead to overfitting. The association constant for the reaction is written as a square, according to $(K_a)^2 = [IA_2]/([I][A]^2)$, where [I] is the concentration of indicator (Ddx4n1-YFP in case 1 and labeled DNA in case 2) and [A] is the concentration of analyte (DNA in case 1 and RP3 in case 2). In the main manuscript, we quote the value of $K_a$, which provides a more intuitive measure for the concentration at which binding occurs than the formal overall binding affinity $(K_a)^2$. The model can be easily modified to make it suitable for even higher stoichiometries. For a 1:3 stoichiometry, the exponents "2" in (Eq. 2) need to be replaced by "3", etc. Supplementary Fig. 11 shows a comparison of fits with the 1:1, 1:2 and 1:3 models to the data for both Ddx4n1-ssDNA and RP₃-ssDNA.

The following parameters were used in the Capflex setup for RP₃-ssDNA binding (Table 6):

**Automated pipetting**. Automatic pipetting of samples (for RP₃ LLPS) with the OT2 robot (OpenTrons, New York City, USA) into V-bottom 96-well plates with seal (FIDAbio, Denmark) was done using a custom Python script using the OpenTrons API. Calculations of sample dilutions were done with inbuilt functions in the FIDA 1 software. The samples were mixed with the last pipetting step in the robot, and the plate was then sealed and transferred to the FIDA 1 instrument for analysis.

**Capflex measurements of droplet formation kinetics of Ddx4n1 in presence of PEG3000**. For the measurements of initial droplet formation kinetics 50 μM Ddx4n1 with 2–4% (w/v) PEG3000 (in 20 mM Tris pH 8.0, 100 mM NaCl, 5 mM TCEP), the samples were incubated in the sample trays at 55 °C, which is above the cloud point. The capillary chamber was kept at 20 °C. A standard capillary with inner diameter of 75 μm and a length of 1 m was used. The following set of instrument parameters were used (Table 7).

Data analysis was performed using origin Pro 2019 (Origin labs, USA), using the asymmetric least squares smoothing baseline with a Symmetric factor of 0.001, threshold of 0.005, smoothing factor of 4 and 20 iterations. The validity of the baseline was verified by visual inspection for all graphs. A local maximum more than 1% different from the baseline is considered a peak and hence a droplet. Averaging peak intensities with a sliding window of 10 peaks then generated overall curves of droplet formation kinetics.

**Relative enrichment factor calculation**. The relative enrichment of protein to DNA inside droplets was determined in a sample of 112 μM Ddx4n1 and a total ssDNA concentration of 3.3 μM. Either Ddx4n1YFP or L-DNA was used as fluorescent reporter. The peaks and curve were integrated using Origin Pro (Origin Labs, USA). Positive signal more than 0.5% from the baseline was considered a peak. The Enrichment factor was calculated by taking the sum of the peak integrals and dividing by the total curve integral in the following way:

$$\text{Ratio} = \sum(\text{peaks})/(\text{Curve area})$$

This ratio for both protein and DNA was then divided, according to:
Relative enrichment = Ratio (Protein)/Ratio (ssDNA)
—Yielding the relative enrichment factor.

**Throughput calculations for Capflex**. Conventional approaches to characterize protein LLPS often require multiple instances of human intervention and supervision and therefore can be person-time consuming. Moreover, the determination of droplet size distributions requires extensive microscopic measurements followed by laborious analysis of the droplet sizes. On the other hand, Capflex can independently be used to measure all the said parameters with little human intervention (person-time) and provides a significant information content with minimal sample consumption. Notably, Capflex can be coupled with OpenTrons OT2 pipetting robot (or, after some additional interfacing work, with any other pipetting robotics system) to further minimize the person-time for sample preparations.

In simple terms, the throughput of a given experimental technique can be described by the following equation (Eq. 3);

$$N = r \times t \tag{3}$$

Where $N$ = number of samples run, $t$ = time required to run $N$ samples and $r$ = rate or throughput of the instrument. Increasing the injection pressure (3500 mbar maximum) and/or using an LLPS system without highly viscous substances (such as PEG) will automatically increase the throughput substantially.

**Microscopy**. Bright field, DIC, and fluorescence microscopy images were recorded at ×40 and ×60 (oil immersion) magnification using an epifluorescence microscope (Nikon Eclipse Ti2 (RAMCON, Denmark)) at the DTU bio-imaging core facility. The images were obtained with a 16-bit depth and with a resolution of 2048 × 2048. An excitation wavelength of 470 nm and emission channel of 500–600 nm was used for all three labeled proteins/peptides used in our study. The exposure time was adjusted accordingly for each sample and image to ensure the highest possible number of droplets are detected. The fusion and droplet re-mixing study for Ddx4n1 and α-Syn LLPS (Supplementary Movies 1–3) were obtained using an inverted bright field microscope (Zeiss Axio vert. A1, (Zeiss, Germany)) with the help of a ×40 objective. The ThT co-partitioning was monitored using an in-built ThT channel in the Zeiss axio vert. A1 microscope. The images were taken in 16-bit depth and with a resolution of 1024 × 1024. All the microscopy images are analyzed subsequently using ImageJ (NIH, USA).

**Droplet size distribution**. Microscopy-based size distribution analysis was performed using a LMI-005-Confocal Microscope-SP8 (Leica Microsystems, Germany) at the DTU bio-imaging core facility. Briefly, 5 μl of 100, 116, 133, and 167 μM of phase-separated Ddx4n1 (spiked with 1 μM YFP Ddx4n1) samples were drop-casted onto a clean glass slide immediately after LLPS at 20 °C. The droplets were imaged using a ×60 oil immersion objective at an excitation wavelength of 470 nm and emission channel of 500–600 nm. The images were acquired with a 16-bit depth and with a resolution of 2048 × 2048. For each image, depth scanning (Z-stack) was performed (40 μm depth scans) to confirm the spherical nature of the droplets. The droplet size was measured for $n$ = 100 droplets for each sample using ImageJ (NIH, USA).

**Fluorescence recovery after photobleaching (FRAP) experiments**. 120 μM Ddx4n1 (spiked with 1 μM YFP Ddx4n1) is phase separated at room temperature (25 °C) and subjected to FRAP experiments with the help of an inverted confocal fluorescence microscope (LMI-005-Leica Microsystems Confocal Microscope). The droplet is bleached at the center with a region of interest (ROI) radius of 2 μm. A 488 nm bleaching laser is used at 100% power. A complete fluorescence recovery is recorded within 20 s. For FRAP of α-Syn liquid droplets, 100 μM α-Syn (spiked with 1 μM Alexa488 α-Syn) is phase separated in the presence of 20% (w/v) PEG6000 and subjected to FRAP experiments at 20 h. The droplet is bleached at the center with a ROI radius of 4 μm. A 488 nm bleaching laser is used at 100% power and complete fluorescence recovery is recorded within 40 s. The normalized fluorescence recovery for both Ddx4n1 and α-Syn is calculated after correction for ROIs with passive bleaching and background fluorescence as per previously

established protocols[47,82,83]. This was done by computing the normalized fluorescence intensity, $I(n)$ using the following equation (Eq. 4);

$$I(n) = \frac{[I(t) - I(b)]}{r} \tag{4}$$

Where, $I(t)$ = fluorescence intensity at time t, $I(b)$ = background fluorescence intensity, $r = I_c/I_{c_0}$ which is the rate of photobleaching, where, $I_{c_0}$ = fluorescence intensity of the given ROI before photobleaching, $I_C$ = fluorescence intensity of the given ROI after photobleaching.

**Cuvette-based spectroscopy and scattering experiments**. Cloud points and the influence of additives on Ddx4n1, effect of additives on Ddx4n1-YFP fluorescence and effect of temperature on α-Syn LLPS was investigated using a multichannel spectrophotometer equipped with a thermal control and IR temperature measurement—the ProbeDrum (ProbationLabs, Lund, Sweden). Samples of 50 μL were scanned from 12 to 60 °C in a high precision Cell Quartz glass cuvette with 3 × 3 mm light path (Hellma Analytics, Germany) while measuring the absorbance at 240–700 nm, fluorescence at 240–700 nm with excitation at 280 and 509 nm as well as static light scattering at 90° with a laser of 636 nm for all time points. The temperature scan rate was 2 °C per minute and measurements were taken every 5 s. Initial data analysis was carried out using PDviewer (ProbationLabs, Sweden) and the data were visualized using Origin Pro 2019 and 2021 (Origin labs, USA) and KaleidaGraph (v4.03).

**From relative peak intensities to relative droplet sizes**. In order to be able to translate a relative Capflex peak intensity distribution into a relative droplet size distribution, the non-uniform flow velocity across the capillary cross-section needs to be considered. Droplets that cross the detection area of the capillary with different flow rates will reside for different time periods in the detection area and hence the integrated signal intensity differs. In addition to the effects of the parabolic flow profile of Poiseuille flow, flow focusing might also need to be considered. Furthermore, potential droplet deformation in the shear field of the capillary flow should also be considered. Below, we discuss these effects separately.

**Radial particle distribution**. Two types of particle focusing mechanisms need to be considered in the FIDA system: Dean Flow and radial migration. Dean flow is a concern when the dean number, $k$ (Eq. 5), is not significantly smaller than one[84].

$$k = \sqrt{\frac{H}{2R}} Re \approx \sqrt{\frac{10^{-4}}{2 \cdot \frac{1}{2}10^{-2}}} 1 = 10^{-1} \tag{5}$$

Where H is the capillary diameter, 'Re' is the Reynolds number of the flow and R is the turn radius of the capillary. Hence, Dean Flow can be neglected in the FIDA 1 setup. Dino Di Carlo has provided some general design rules for particle focusing microfluidic systems[84], which will be utilized to generalize whether radial migration is a concern in this system. The conditions explored here are the most advantageous conditions for particle focusing for this study.

The Hagen-Poiseuille equation can be used to predict the velocity of the fluid in a capillary (Eq. 6)[85]:

$$U_m = \frac{\triangle P\,D^2}{4\mu L} = \frac{10^5\,10^{-8}}{8\,10^{-3}}\frac{m}{s} \approx 10^{-1}\frac{m}{s} \tag{6}$$

Where $U_m$ is maximum velocity, $\Delta P$ is the pressure, D is the capillary diameter, μ is dynamic viscosity (approximated to that of pure water) and L is capillary length.

The length necessary for significant focusing of general particles/droplets is expressed as (Eq. 7):

$$L_f = \frac{\pi\mu H^2}{\rho U_m a^2 f_e} = \frac{\pi 10^{-3} 10^{-8}}{10^3 \cdot 10^{-1} \cdot 10^{-12} \cdot 0.05}m \approx 60\frac{10^{-11}}{10^{-10}}m = 6m \tag{7}$$

Where H is smallest capillary dimension (diameter), ρ is fluid density (approximated to that of pure water), $a$ is particle radius and $f_e$ is a lift force factor usually between 0.02 and 0.05. Since significant radial migration is only obtained after several capillary lengths in the highest velocity case, it is safe to assume that particles in this experiment are randomly distributed throughout the cross-section, assuming random nucleation of droplets.

**Signal intensity distribution from randomly distributed particles**. Assuming a random particle distribution across the cross-section of the capillary, it is possible to evaluate a signal profile of a monodisperse particle solution. Particle flux throughout the radius must be proportional to flow velocity at any radius multiplied by the probability of finding a particle at any radius. The probability expression must be proportional to the circumference and hence linearly proportional to the radius (Supplementary Fig. 1).

$$\text{particle flux} \propto \left(1 - \frac{r}{R}\right)^2 \cdot r$$

This distribution has been numerically correlated with signal intensity, which is related to the reciprocal particle velocity, and the resulting signal distribution has

then been fitted to the calibration data acquired with the 1 μm polystyrene monodisperse spheres (Supplementary Fig. 2).

The model reproduces the experimentally observed tail at higher signal intensities well, supporting the assumption of random particle distribution described above. Several signal intensities of significantly lower than the expected minimal signal (corresponding to particles in the center of the capillary traveling at the fastest flow rate) are likely to be explained by the intrinsic variability of the absolute fluorescence intensity of the particles (Supplementary Fig. 4).

**Shear stress and droplet deformation**. LLPS droplets are likely to undergo some degree of deformation, when exposed to the shear stress of a laminar flow. The maximum expected shear stress during these experiments is at the inner wall of the capillary (Eq. 8):

$$\tau_w = \gamma_w \cdot \mu = \frac{8U_m}{D}\mu \approx \frac{2 \cdot 10^{-1}}{10^{-4}}10^{-3}Pa = 2Pa \tag{8}$$

Which is similar to the stress which has been found to deform and induce aggregation in FUS droplets[86]. However, in these experiments, a channel geometry optimized for droplet deformation was used (perpendicular channels).

Quantifying the degree and effect of shear stress on droplet deformation and potential breakup is difficult, as very little is known about the physical characteristics of such droplets (surface tension, viscosity) and their dependence on solution conditions. While droplet deformation will impact their flow through the capillary, it is unlikely to affect the detection and quantification of droplets. The total fluorescence intensity emitted by a droplet is likely to be unaffected by its shape, as well as by a potential shear-mediated liquid-to-solid transition, as the fluorophore quantity of each droplet is maintained throughout such a transition. It should be noted that there is currently no evidence that the protein system used in the present study (Ddx4n1) is able to undergo such a shear-mediated liquid-to-solid transition. At very large shear forces, droplet breakup can influence the quantification of relative droplet size distributions. However, most Capflex experiments were performed at moderate flow rates and no evidence for droplet breakup was observed under any conditions.

**Amino acid sequences**. TEV Protease: [MBPTEVsiteHISTEVprotease]

[MFNLQEPYFTWPLIAADGGYAFKYENGKYDIKDVGVDNAGAKAGLTFL VDLIKNKHMNADTDYSIAEAAFNKGETAMTINGPWAWSNIDTSKVNYGVT VLPTFKGQPSKPFVGVLSAGINAASPNKELAKEFLENYLLTDEGLEAVNKDKP LGAVALKSYEEELAKDPRIAATMENAQKGEIMPNIPQMSAFWYAVRTAVINA ASGRQTVDEALKDAQTNSSSNNNNNNNNNNLGIEGRGENLYFQGHHHHH HHGESLFKGPRDYNPISSTICHLTNESDGHTTSLYGIGFGPFIITNKHLFRRNNG TLLVQSLHGVFKVKNTTTLQQHLIDGRDMIIRMPKDFPPFPQKLKFREPQRE ERICLVTTNFQTKSMSSMVSDTSCTFPSSDGIFWKHWIQTKDGQCGSPLVSTR DGFIVGIHSASNFTNTNNYFTSVPKNFMELLTNQEAQQWVSGWRLNADSVL WGGHKVFMVKPEEPFQPVKEATQLMNRRRRR]

GST-TEVsite-Ddx4n1: [HisGSTTEVsiteDdx4n1]

[HHHHHHMSPILGYWKIKGLVQPTRLLLEYLEEKYEEHLYERDEGDKWRN KKFELGLEFPNLPYYIDGDVKLTQSMAIIRYIADKHNMLGGCPKERAEISMLE GAVLDIRYGVSRIAYSKDFETLKVDFLSKLPEMLKMFEDRLCHKTYLNGDHVT HPDFMLYDALDVVLYMDPMCLDAFPKLVCFKKRIEAIPQIDKYLKSSKYIAW PLQGWQATFGGGDHPPKSDLVPRGSPGIHRDENLYFQGGAMGSMGDEDWE AEINPHMSSYVPIFEKDRYSGENGDNFNRTPASSSEMDDGPSRRDHFMKSGFA SGRNFGNRDAGECNKRDNTSTMGGFGVGKSFGNRGFSNSRFEDGDSSGFWR ESSNDCEDNPTRNRGFSKRGGYRDGNNSEASGPYRRGGRGSFRGCRGGFGLG SPNNDLDPDECMQRTGGLFGSRRPVLSGTGNGDTSQSRSGSGSERGGYKGLN EEVITGSGKNSWKSEAEGGES]

His_tag-GST-TEV_site-Ddx4n1-YFP: [HisGSTTEVsiteDdx4n1YFP]

[HHHHHHMSPILGYWKIKGLVQPTRLLLEYLEEKYEEHLYERDEGDKWRN KKFELGLEFPNLPYYIDGDVKLTQSMAIIRYIADKHNMLGGCPKERAEISMLE GAVLDIRYGVSRIAYSKDFETLKVDFLSKLPEMLKMFEDRLCHKTYLNGDHV THPDFMLYDALDVVLYMDPMCLDAFPKLVCFKKRIEAIPQIDKYLKSSKYIA WPLQGWQATFGGGDHPPKSDLVPRGSPGIHRDENLYFQGGAMGSNMGDE DWEAEINPHMSSYVPIFEKDRYSGENGDNFNRTPASSSEMDDGPSRRDHFMK SGFASGRNFGNRDAGECNKRDNTSTMGGFGVGKSFGNRGFSNSRFEDGDSSG FWRESSNDCEDNPTRNRGFSKRGGYRDGNNSEASGPYRRGGRGSFRGCRGG FGLGSPNNDLDPDECMQRTGGLFGSRRPVLSGTGNGDTSQSRSGSGSERGGY KGLNEEVITGSGKNSWKSEAEGGESSDTQGPKVTLQMVSKGEELFTGVVPILV ELDGDVNGHKFSVSGEGEGDATYGKLTLKFICTTGKLPVPWPTLVTTFGYGL MCFARYPDHMKQHDFFKSAMPEGYVQERTIFFKDDGNYKTRAEVKFEGDTL VNRIELKGIDFKEDGNILGHKLEYNYNSHNVYIMADKQKNGIKVNFKIRHNI EDGSVQLADHYQQNTPIGDGPVLLPDNHYLSYQSKLSKDPNEKRDHMVLLEF VTAAGIT]

Human α-Synuclein [SNCA]

MDVFMKGLSKAKEGVVAAAEKTKQGVAEAAGKTKEGVLYVGSKTKEGV VHGVATVAEKTKEQVTNVGGAVVTGVTAVAQKTVEGAGSIAAATGFVKK DQLGKNEEGAPQEGILEDMPVDPDNEAYEMPSEEGYQDYEPEA

A140C α-Synuclein

MDVFMKGLSKAKEGVVAAAEKTKQGVAEAAGKTKEGVLYVGSKTKEGV VHGVATVAEKTKEQVTNVGGAVVTGVTAVAQKTVEGAGSIAAATGFVKK DQLGKNEEGAPQEGILEDMPVDPDNEAYEMPSEEGYQDYEPEC

RP3 peptide
RRASL RRASL RRASL

**Reporting summary**. Further information on research design is available in the Nature Research Reporting Summary linked to this article.

## Data availability

The authors declare that all the data supporting the findings of this study are available within the paper and in supplementary information files. All the data analysis was performed using published tools and packages and has been cited in the paper and supplementary information text. No data has been excluded. Source data are provided with this paper.

## Code availability

Baseline and peak height determination of Capflex measurements was carried out with a python script available here: https://doi.org/10.11583/DTU.14223116. OpenTrons OT2 automated pipetting code: This code can be found using the https://doi.org/10.11583/DTU.16797556.

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

## Acknowledgements

Charlotte O'Shea from University of Copenhagen is thanked for donating the TEV-MBP plasmid. We thank Dr. Tim Nott (Department of Biochemistry) and Dr. Andrew Baldwin (Department of Chemistry), both at Oxford University, for donating the PET-30M plasmids containing the GST-Ddx4n1 fusion constructs and for help with protein production. The DTU bio-imaging core (at DTU bioengineering) is acknowledged for confocal imaging and FRAP experiments. Kristina Mielec is thanked for technical assistance. E.G.P.S, R.K.N, J.A.L., and A.K.B. would like to acknowledge the Novo Nordisk Foundation for funding (Grant number: NNFSA170028392). S.R. and A.K.B. would like to acknowledge VILLUM FONDEN for financial support (Grant number 35823). C.G. thanks the Lundbeck Foundation (Grant number: R314-2018-3493) and C.G. and D.P. thank the Carlsberg Foundation (Grant number: CF19-0382). The Carlsberg Foundation is also acknowledged for funding the FIDA 1 instrument with a grant to Andreas H. Lausten (Grant number: CF19-0055). Funding from Novo Nordisk Foundation (grant NNFOC0055625) for the infrastructure "Imaging microbial language in biocontrol (IMLiB)" is acknowledged. We would like to thank Lars Boyens-Thiele for assistance with experiments.

## Author contributions

E.G.P.S., S.R., R.K.N., J.A.L., and A.F. designed and performed experiments and analyzed data. H.J. designed experiments and analyzed data. D.P. and C.G. contributed reagents and methods. A.K.B. conceived the study, designed experiments and analyzed data. E.G.P.S., S.R., and A.K.B wrote the manuscript. S.R. prepared all schematics and illustrations. All authors commented on the manuscript.

## Competing interests

H.J. is the C.S.O. of FIDA Biosystems. The authors declare no competing interests.
