## [Peer Review File · Nature Communications]

Reviewers' Comments:

Reviewer #1:

Remarks to the Author:

This paper describes a very promising method to characterize several properties of biomolecular condensates with high-throughput, i.e. thermodynamics, kinetics, role of modifiers, liquid to fibril transition. Several aspects of this instrument are particularly strong: i) the possibility of automation, therefore allowing high throughput; ii) the ability to change temperature between reservoir and the chamber of analysis; iii) the accessibility to essentially any user. Here the authors demonstrate the potential of this technique by analysing three different systems of high significance. The paper is clearly written. This method has great potential and will be of high relevance for the field. I envision applications in many different systems. I therefore recommend acceptance after addressing the points below.

General considerations:

1) As the authors mentioned, the information obtained with this instrument can be achieved with other methods. One of the major strengths and unique point of this instrument is the automation, which allows high throughput. The results however could highlight more this feature. For instance the authors could comment on how they exploited this feature in this study, i.e. which unique benefits were obtained thanks to the high-throughput. Moreover, the throughput should be quantified by describing how many samples/conditions can be tested e.g. in one day, and compared with more classical methods.

2) Connected to the previous point, the throughput of the technique would be incredibly useful for screening campaigns, where presence/absence of LLPS may not be known a priori in different solution conditions. In this case, since the detection of the method is based on spikes in the fluorescence intensity, a challenge is to differentiate LLPS from other types of phase separation (e.g. aggregation, precipitation). An example along this line is provided with aSyn fibrils, which are ThT positive. The authors may add some additional comments on this potential application of the technique (which I believe will be very useful in saving a lot of time and effort).

Specific points:

3) Figure 1: Microscopy images of the condensates show spherical shapes (Fig. 1 h) but additional evidence of liquidity (e.g. fusion events, FRAP) is required, as nicely done for a-Syn in Fig. 5

4) Figure 1d, line 164 "Our data showed that ... concentrations of dilute phase and dense phase..." To my understanding the method can measure the concentration in the dilute phase (the critical concentration), not the concentration in the dense phase. I also found a bit difficult to read Fig. 1d. It is up to the authors but my suggestion would be to split this panel in two, or modify it.

5) Effect of CaCl₂: a control with MgCl₂ could be performed to show the specific effect of Ca.

6) Figure 1 caption (and other Figures): "with similar observations": this statement should be quantified or independent experiments should be shown

7) Size distribution: size distributions of droplets at different initial protein concentrations can be estimated by bulk microscopy images and (at least qualitatively) compared with the distribution of the fluorescence intensity (direct quantitative comparison may be hindered by sedimentation and wetting in the bulk assay).

8) Affinity Ddx4-ssDNA: Stoichiometry of binding to individual protein molecules shows that more than a single DNA molecule interacts with one molecule of Ddx4, with low affinity. However, highly sub-stoichiometric concentrations of ssDNA dissolve the droplets. In the conclusions this is explained with a higher affinity (one order of magnitude) of ssDNA for the droplet state. How was the affinity for the droplet phase calculated? Wouldn't a different affinity of ssDNA for continuous and dense phase lead to a different stoichiometry in the droplets with respect to the overall stoichiometry (in contrast with what has been measured?).

9) Fig. 3d: "possible transition from binodal to spinodal": this statement is just briefly mentioned in the caption and is difficult to follow. More explanation is required in the main text

10) Introduction: as current techniques to measure phase diagrams, NMR could also be mentioned although requiring large amount of sample (Brady J.P. et al., PNAS, 2017, 114, 39) and fluorescence correlation spectroscopy (Wei M.T. et al., Nature Chemistry, 1118-1125, 2017). For microfluidic approaches, in addition to ref. 52, please consider also Bremer A. et al., Lab Chip, 2020, 20, 4225-4234 and Kopp M. et al, 2020, 92, 8, 5803; instead or in addition to ref. 54 please

consider Kueffner et al, ChemSystemsChem, 2020, 2, e2000001, where Raman spectroscopy was used earlier to measure protein concentration inside individual droplets.

Reviewer #2:

Remarks to the Author:

The manuscript by Stender et al. entitled "Capillary flow experiments (Capflex) for thermodynamic and kinetic characterization of protein liquid-liquid phase separation at high throughput" describes a highly novel study that will make a significant impact in the field of liquid-liquid phase separation (LLPS). The authors utilize an automated Capillary Flow Experiment (abbreviated as Capflex) method that is based on Flow-Induced Dispersion Analysis (FIDA). This unique application of FIDA allowed the authors to obtain a wealth of highly quantitative information during LLPS of several unrelated intrinsically disordered proteins/peptides with or without nucleic acids. The authors demonstrated the power of FIDA-based Capflex in the quantitative estimation of the dilute phase concentration, droplet size distributions, and kinetics of LLPS and LLPS-mediated amyloid formation. Using this methodology, the authors were able to characterize the LLPS of Ddx4n1, complex coacervation of ssDNA and RP3, and LLPS/LLPS-mediated liquid-to-solid phase transition of alpha-synuclein into amyloids. This study represents a technological tour de force and is of great interest in quantitative in vitro characterizations of biological phase separation. This work is an important contribution and is accessible to a broad audience, and therefore, is suitable for publication in Nature Communications. Below I summarize my comments that can be addressed before this work is accepted for publication.

1. The authors state in lines # 163-165 "Our data showed the absolute concentrations of dilute phase and dense phase do not change with total protein concentration, only their relative volume fractions (Figure 1d)". They estimate the light phase concentration from the baseline fluorescence signal (using a standard curve calibration). How do they obtain the dense phase concentration? The dense phase concentration estimation might be difficult using this method because the peak intensities (or integral intensities) of the spikes (or bursts) are highly convoluted due to the dense phase concentration, a wide droplet size distribution, and bias due to the laminar flow (as the authors stated in lines 210-212: "rapid flow in the center of the capillary or slower flow at the edges"). Can the authors comment on the approximate dense phase concentration? I am not sure if some form of autocorrelation analysis can help determine the approximate dense phase concentration. If the dense phase concentration cannot be estimated by the Capflex method, I was wondering if the authors could consider stating this limitation.

2. It's a bit hard to follow Figure 1d. Authors can consider simplifying this figure to clearly show the dependence on PEG, CaCl₂, etc. The unit is missing for the total protein concentration on the right ordinate (Y-axis).

3. Kinetics of Ddx4n1 LLPS. These results are very interesting (Figure 3). The authors directly show an acceleration of LLPS as a function % of PEG. Does the delay they observe at a lower % of PEG (2%) indicate a nucleation phenomenon? It's a bit confusing whether this 30 sec delay is partly due to the deadtime of their observation. The deadtime (~ 15 sec) is not clearly mentioned in the manuscript and stated in lines 307-308 and at the end in the discussion section (lines 549-550). The question is whether there is a nucleation process that exhibits sigmoidal kinetics or this delay (lag time) is related to the deadtime. Of course, in fast LLPS kinetic traces (3-5% PEG), this delay is shortened but not completely eliminated. And when the authors investigated pre-formed amyloid (LLPS-mediated alpha-synuclein amyloids) (Figure 5), they also observe ~2-3 min delay. It would be helpful for the readers if the authors could clarify the observed delays in their kinetic traces.

4. Figure 3 legend (lines 341-342). "The drop in the peak heights after 1.5 min represents possible transition from a binodal LLPS to spinodal decomposition". This aspect is unclear and has not been explained in the manuscript text. Also, they state "The solid lines indicate the average increase in the peak intensity for respective conditions." It's not clear whether the solid lines are for guiding the eyes. The authors could consider stating this clearly.

5. Methods. Since this is primarily a Method manuscript and most readers may not be familiar with FIDA, I was wondering the authors could briefly state this methodology with some essential technical details including deadtime in a Methods section (after Discussion). Other experimental details can be in Supplementary Information. Also, the authors could consider citing an earlier/introductory paper on FIDA (J. Am. Chem. Soc. 2010, 132, 4070-4071 in which one of the authors of the current manuscript was an author).

6. Journal names are missing in some references (For example, Ref. 48-50).

Reviewed by Samrat Mukhopadhyay

Reviewer #3:

Remarks to the Author:

I currently have a split role in Industry and Academia. The work here is highly original and the ability to perform capillary high throughput analysis for quantitative IIPs is highly relevant and exciting to drug discovery activities in University and Industry. The multiple case studies presented amply support the conclusions and claims. I identified no flaws in the interpretation of this sound piece of work. The methods are provided in sufficient detail. I expect this approach will find wide interest to those working in drug discovery. I very much enjoyed reading the manuscript which was easily understandable.

Reviewer #1 (Remarks to the Author):

This paper describes a very promising method to characterize several properties of biomolecular condensates with high-throughput, i.e. thermodynamics, kinetics, role of modifiers, liquid to fibril transition. Several aspects of this instrument are particularly strong: i) the possibility of automation, therefore allowing high throughput; ii) the ability to change temperature between reservoir and the chamber of analysis; iii) the accessibility to essentially any user. Here the authors demonstrate the potential of this technique by analyzing three different systems of high significance. The paper is clearly written. This method has a great potential and will be of high relevance for the field. I envision applications in many different systems. I therefore recommend acceptance after addressing the points below.

Response: We thank the reviewer for his/her positive comments on our work and for judging it suitable for publication in *Nature Communications*. We are grateful for your comments, which will significantly improve the quality of the current study.

General considerations:

1) As the authors mentioned, the information obtained with this instrument can be achieved with other methods. One of the major strengths and unique point of this instrument is the automation, which allows high throughput. The results however could highlight more this feature. For instance, the authors could comment on how they exploited this feature in this study, i.e. which unique benefits were obtained thanks to the high-throughput. Moreover, the throughput should be quantified by describing how many samples/conditions can be tested e.g. in one day, and compared with more classical methods.

Response: We thank the reviewer for raising this point. We agree that the information obtained from Capflex experiments using FIDA 1 (dilute phase concentration, droplet size distribution and droplet formation kinetics) can be achieved using a number of other methods such as DIC and fluorescence microscopy¹, fluorescence correlation spectroscopy², microfluidics³⁻⁵, quantitative phase microscopy⁶ and Raman microscopy⁷. NMR can also be used to decipher protein phase diagrams although requiring a large amount of sample⁸. These techniques are frequently used in combination with additional experimental steps such as centrifugation and UV absorption spectroscopy/scattering measurements⁹. Conventional approaches also often require multiple instances of human intervention and supervision and therefore can be person-time consuming. Moreover, the determination of droplet size distributions requires extensive microscopic measurements followed by laborious analysis of the droplet sizes. On the other hand, Capflex can independently be used to measure all the said parameters with little human intervention (person-time) and provides a significant information content with minimal sample consumption. Notably, Capflex can be coupled with OpenTrons OT2 pipetting robot (or, after some additional interfacing work, with any other pipetting robotics system) to further minimize the person-time for sample preparations. These benefits of Capflex are now described in the revised manuscript.

After the reviewer's suggestions, we have realized that a more thorough quantification of the throughput is indeed valuable for the readers of our work. In simple terms, the throughput of a given experimental technique can be described by the following equation;

$$N = r \times t$$

Where N = number of samples run, t = time required to run N samples and r = rate or throughput of the instrument.

The FIDA1 instrument can run two 96 well plates simultaneously. LLPS systems that require high temperature incubation (such as Ddx4n1 where the tray temperature is kept at 50 °C to prevent LLPS in the tray) might lead

to sample evaporation during prolonged campaigns with the current setup of the instrument. Therefore, to demonstrate and calculate the throughput of Capflex, we have chosen α -Syn LLPS. We have analyzed LLPS behavior of α -Syn at a range of protein and PEG8000 concentrations at 37 °C. PEG8000 was chosen since the increased crowding effect (compared to PEG6000) induces instantaneous LLPS of α -Syn at high micromolar range¹⁰.

We used 20, 50, 100, 150 and 200 μ M α -Syn (in 20 mM PBS (commercially available tablets, Sigma, USA), pH 7.4) in presence of increasing (0, 10 and 20% (w/v) PEG8000 for an orthogonal screen of α -Syn LLPS (**Figure R1** of this letter and **Supplementary Figure 13** of the revised manuscript). The samples were prepared using the OpenTrons OT2 pipetting robot. The protein, buffer and PEG stocks were prepared manually. Subsequent experimental steps required no manual interventions. The sample preparation time by the OpenTrons OT2 robot was 36 min. Next, the samples were directly loaded in the FIDA1. For 15 samples used in our study, the FIDA1 run-time was measured as 180 min with a sample injection pressure of 2000 mbar. The measured throughput was calculated as $r = 15 \text{ samples} / 3.6 \text{ h} = \sim 4 \text{ samples} / \text{h} = 96 \text{ samples} / \text{day}$ including sample preparation time with no human supervision. Each sample vial contained 40 μ l of sample for two iterations/sample. Therefore, the total sample volume to run this orthogonal LLPS screen was only 0.6 ml. Nevertheless, increasing the injection pressure (3500 mbar maximum) and/or using an LLPS system without highly viscous substances (such as PEG) will automatically increase the throughput substantially.

Our results showed instantaneous (0 h) α -Syn LLPS at 200 μ M concentrations in presence of 10 and 20% (w/v) PEG8000 (**Figure R1** of this letter and **Supplementary Figure 13** of the revised manuscript) with a partitioning of $\sim 30 \mu$ M dilute phase equivalent α -Syn in the droplet phase. We have demonstrated this by plotting the [total protein concentration – dilute phase concentration] (**Figure R1** of this letter and **Supplementary Figure 13** of the revised manuscript). The non-phase separated samples showed $\pm 7 \mu$ M difference in the total protein concentration, which was within the range of error. Together, our results demonstrates an individual campaign at HTP with minimal sample consumption and human supervision. This data and the respective discussion is now incorporated under the paragraph named ‘*Capflex throughput characterization*’ in the revised version of our manuscript. Additionally, from this experiment, we note that the technique can also provide non-binary phase diagrams by measuring the dilute phase concentration.

Figure R1: Samples of 20, 50, 100, 150 and, 200 μ M α -Syn (in 20 mM PBS, pH 7.4) in the presence of increasing (0, 10 and 20% (w/v) PEG8000 concentration are analyzed with Capflex. The solutions are spiked with 50 nM Alexa488 maleimide labeled A140C α -Syn. 200 μ M α -Syn in the presence of 10 and 20% (w/v) PEG8000 shows clear signs of LLPS as confirmed by the appearance of spikes in the fluorescence signal and a substantial decrease in the baseline fluorescence signal (shown as insets). The mean dilute phase equivalent concentration is represented with color codes (right) and quantified from the baseline fluorescence intensities for individual samples (represented as circles) for $n=2$ observations. Representative Capflex traces are shown. The traces are indicated in black when there is no LLPS and green when there is successful LLPS.

2) Connected to the previous point, the throughput of the technique would be incredibly useful for screening campaigns, where presence/absence of LLPS may not be known a priori in different solution conditions. In this case, since the detection of the method is based on spikes in the fluorescence intensity, a challenge is to differentiate LLPS from other types of phase separation (e.g. aggregation, precipitation). An example along this line is provided with α -Syn fibrils, which are ThT positive. The authors may add some additional comments on this potential application of the technique (which I believe will be very useful in saving a lot of time and effort).

Response: The reviewer is correct that similar to true liquid droplets, solidified droplets and/or small aggregates will also show fluorescence intensity spikes in the FIDA 1 instrument. For temperature sensitive LLPS samples (such as Ddx4n1), liquid droplets will spontaneously re-dissolve above the cloud point leading to disappearance of fluorescence signal spikes. Solid aggregates, if present, will persist in the solution and will manifest as spikes even after the solution temperature has been raised to above the cloud point.

Using this concept, we have now analyzed the ability of Ddx4n1 liquid droplets to dissolve upon heating. To do this, 140 μ M of Ddx4n1 in 20 mM Tris, pH 8.0, 100 mM NaCl, 5 mM TCEP was spiked with 500 nM of

YFP labeled Ddx4n1 and subjected to Capflex measurements. The sample tray temperature was kept at 15 °C (below the cloud point) to induce LLPS of Ddx4n1. The capillary temperature was also kept at 15 °C. Our data shows successful phase separation as evident from the fluorescence signal spikes in the FIDA 1 (**Figure R2** of this letter and **Supplementary Figure 4** of the revised manuscript). Subsequently, both the sample tray and capillary temperature was increased to 50 °C (above the cloud point). At 50 °C, no fluorescence signal spikes were observed from the same sample indicating re-dissolution of the droplets in the solution above cloud point (**Figure R2** of this letter and **Supplementary Figure 4** of the revised manuscript). The baseline at 50 °C was decreased because the fluorescence quantum yield of YFP is sensitive towards temperature. Our results clearly showed that Ddx4n1 signal spikes arise from completely reversible, phase separated liquid droplets. The reversible nature of Ddx4n1 droplets was supported by temperature dependent light scattering measurements (in presence of increasing PEG3000 concentration) in the ProbeDrum (**Supplementary Figure 3** of the revised manuscript).

Figure R2: Ddx4n1 liquid droplets dissolve above cloud point temperature in Capflex

Appearance of spikes in the fluorescence signal confirms LLPS of 140 μM Ddx4n1 (500 nM YFP Ddx4n1 as reporter) at 15 °C (blue trace). The same sample when heated to 50 °C shows no signal spikes confirming the reversible nature of liquid Ddx4n1 droplets. Both the sample tray and capillary temperatures are kept at identical temperatures during each measurement. Representative Capflex traces are reported. The experiment was performed two times with similar observations.

A more suitable protein system to differentiate between liquid droplets and aggregates using Capflex is α -Syn. This is because LLPS of α -Syn precedes its aggregation^{10,11}. However, α -Syn LLPS is not very temperature sensitive in the range that can be used in FIDA 1 (10-55 °C). In such systems, dilution below the critical concentration of the sample would lead to re-mixing of the phase separated droplets. We had characterized this phenomenon in the originally submitted version of the manuscript where we showed that at 24 h, α -Syn droplets were reversible and re-dissolved upon dilution—leading to disappearance of fluorescence spikes in the FIDA 1 (**Figure 5h** of the revised manuscript).

We have now extended these initial experiments and have probed the kinetics of α -Syn LLPS from a reversible to a non-reversible state in a more highly resolved manner using Capflex (**Figure R3** of this letter and **Figure 6** of the revised manuscript). To do this, 100 μM of α -Syn + 20% (w/v) PEG6000 (in 10 mM sodium phosphate buffer saline (PBS) ($\text{Na}_2\text{HPO}_4 + \text{NaH}_2\text{PO}_4$), 200 mM NaCl, pH 7.4) was incubated at 37 °C to induce LLPS. We used three sets of samples to probe LLPS and the liquid-to-solid transition of α -Syn using Capflex. Please note that all the sets were prepared from one single master stock solution.

a) The first set of samples were spiked with 10 nM of Alexa488 labeled α -Syn and Capflex measurements were performed at an interval of 4 h (during 48 h). At each time-point, the sample was diluted with buffer and the reversibility (disappearance of signal spikes upon re-mixing) of the droplets was probed using Capflex (**Figure R3** of this letter and **Figure 6** of the revised manuscript).

b) The second set of samples were spiked with 5 μ M of ThT and recorded at different time-points (for 48 h) to monitor amyloid aggregation in Capflex. (**Figure R3** of this letter and **Figure 6** of the revised manuscript)

c) It is well established that α -Syn amyloid aggregation (monitored using ThT) follows a sigmoidal growth kinetics with three distinct phases—the lag phase, the exponential phase and the saturation or plateau phase. To correlate the extent of reversibility of α -Syn droplets and the dilute phase concentration with the aggregation kinetics of α -Syn, the third set of samples were incubated with 10 μ M of ThT and the aggregation kinetics was recorded with a plate reader in parallel to the Capflex measurements (**Figure R3** of this letter and **Figure 6** of the revised manuscript).

Our plate reader aggregation assay using ThT showed that α -Syn LLPS solution (100 μ M of α -Syn + 20% (w/v) PEG6000 has a lag time of ~20-24 h. The exponential aggregation phase starts after ~20-24 h of incubation and continues until ~50 h. The system reaches saturation afterwards (**Figure R3a** of this letter and **Figure 6** of the revised manuscript).

In Capflex, at 0 h, no peaks were observed in the Alexa channel with a stable baseline (at 100 μ M) indicating that the sample was not phase separated (**Figure R3b left panel** of this letter and **Figure 6** of the revised manuscript). Our observation showed that after 4-8 h of incubation, small and sparse peaks started to appear in the Capflex traces, which we speculate could be due to possible nucleation events for α -Syn LLPS. The dilute phase concentrations also decreased slightly down to ~97-98 μ M (**Figure R3b, middle panel** of this letter and **Figure 6** of the revised manuscript). However, microscopic observations showed no detectable droplets until 8 h probably due to the rare occurrence and small size of the droplets (data not shown). After 16 h, α -Syn phase separates into liquid droplets and the dilute phase concentration decreases to ~70 μ M and remains almost unchanged until 24 h (**Figure R3b, left panel** of this letter and **Figure 6** of the revised manuscript). The signal spikes completely disappeared when the phase-separated samples at 16, 20 and 24 h were subjected to dilution suggesting the liquid-like, reversible nature of the droplets (**Figure R3b, right panel** of this letter and **Figure 6** of the revised manuscript). In the ThT channel, we observed no peaks for 0 and 10 h indicating no amyloid fibril formation. The 24 h samples showed few ThT positive peaks, which could be possibly due to the initiation of amyloid aggregation (**Figure R3c** of this letter and **Figure 6** of the revised manuscript).

During the exponential phase of aggregation (30-50 h), the baseline fluorescence in the Alexa488 channel decreased significantly to ~40-50 μ M with substantially more signal spikes—indicating more protein had partitioned into the dense phase (**Figure R3d left and middle panel** of this letter and **Figure 6** of the revised manuscript). Interestingly, the signal spikes did not disappear even after diluting the sample (**Figure R3d right panel** of this letter and **Figure 6** of the revised manuscript). After 48 h, the baseline fluorescence reached ~2-5 μ M, suggesting an irreversible liquid-to-solid transition and aggregation of α -Syn in the exponential phase. During this time (30-48 h), the baseline concentration increased substantially in the ThT channel with signal spikes—suggesting the presence of ThT positive, amyloid aggregates (**Figure R3e** of this letter and **Figure 6** of the revised manuscript).

Together, our data shows that α -Syn LLPS initiates at the lag phase of aggregation where it is completely reversible. The dilute phase concentration remains stable for ~8 h after LLPS when the droplets grow by fusion and ripening. However, irreversible liquid-to-solid phase transition and amyloid aggregation happens primarily during the exponential phase (**Figure R4** of this letter and **Figure 6** of the revised manuscript) as confirmed

by further decrease in the dilute phase concentration and ThT positive signal spikes after 30 h. These extended results are now incorporated in the revised version of the manuscript.

Figure R3: Simultaneous monitoring of $\alpha\text{-Syn LLPS}$ and subsequent amyloid aggregation using Capflex

a. Normalized ThT fluorescence showing amyloid fibril formation of 100 μM $\alpha\text{-Syn}$ (in 10 mM PBS, 200 mM NaCl, pH 7.4) in the presence of 20 % (w/v) PEG6000. The lag, exponential and saturation phase are delineated with black dashed lines. The experiment was performed two times with similar results. **b. (Left panel)** Capflex measurement of $\alpha\text{-Syn LLPS}$ sample spiked with 10 nM Alexa488 labeled $\alpha\text{-Syn}$ until 24 h is shown. With time, the sample shows the appearance of peaks indicative of LLPS by $\alpha\text{-Syn}$. **(Middle panel)** A magnified view of the fluorescence readouts from Capflex showing the evolution of dilute phase concentration during $\alpha\text{-Syn LLPS}$. The baseline fluorescence decreases slightly with small peaks after 4 h of incubation. From 16-24 h, the dilute phase concentration is calculated as $\sim 70 \mu\text{M}$. **(Right panel)** until 24 h, the reversible liquid droplets readily dissolve upon dilution with buffer as confirmed by the disappearance of signal spikes in Capflex. The data is obtained from the same samples that are recorded in the **left** and **middle** panel. **c.** No ThT signal spikes are observed for 0 and 10 h samples. However, after 24 h, small signal spikes are visible which could be due to initiation of amyloid aggregation. **d. (Left panel)** Capflex measurements of $\alpha\text{-Syn LLPS}$ samples spiked with

10 nM Alexa488 maleimide labeled A140C α -Syn from 30-48 h is shown. (**Middle panel**) A magnified view of the fluorescence readouts from Capflex showing the dilute phase concentrations during 30-48 h incubation. The baseline fluorescence further decreases and the dilute phase concentration is calculated as ~ 40 - $50 \mu\text{M}$ during this time. (**Right panel**) The droplet signal spikes do not dissolve upon dilution with buffer indicating irreversible nature of the system after 30 h. **e.** ThT signal spikes are observed for the samples after 30 h suggesting the presence of amyloid aggregates. Representative Capflex traces are reported. The experiment was performed two times with similar results.

Figure R4: Reversible-irreversible phase transition of α -Syn LLPS demonstrated by Capflex

Normalized ThT fluorescence showing amyloid aggregation of $100 \mu\text{M}$ α -Syn (in 10 mM PBS, 200 mM NaCl, pH 7.4) in the presence of 20% (w/v) PEG6000 (LLPS solution) (**red**) is overlaid with the dilute phase concentration obtained from Capflex measurements (**green**). The lag, exponential and saturation phases are delineated with black dashed lines. (Below) Schematic depicting the aberrant phase transition of α -Syn LLPS from a reversible state with $\sim 70 \mu\text{M}$ dilute phase to non-reversible state with $\sim 40 \mu\text{M}$ dilute phase. Eventually, the dilute phase concentration decreases substantially down to $\sim 5 \mu\text{M}$ at the saturation phase of aggregation.

Specific points:

3) Figure 1: Microscopy images of the condensates show spherical shapes (Fig. 1 h) but additional evidence of liquidity (e.g. fusion events, FRAP) is required, as nicely done for a-Syn in Fig. 5

Response: We thank the reviewer for this suggestion. We have now incorporated FRAP of Ddx4n1 liquid droplets along with fusion events to probe their liquid-like nature (**Supplementary Video 1** of the revised manuscript). For FRAP, $120 \mu\text{M}$ Ddx4n1 (spiked with $1 \mu\text{M}$ YFP Ddx4n1) is phase separated at room temperature ($25 \text{ }^\circ\text{C}$) and subjected to fluorescence recovery after photobleaching (FRAP) experiments with the help of an inverted confocal fluorescence microscope (LMI-005-Leica Microsystems Confocal Microscope). The droplet is bleached at the center with a ROI radius of $2 \mu\text{m}$. A 488 nm bleaching laser is used at 100% power. A complete fluorescence recovery is recorded within 20 s . The revised datasets are now incorporated in **Figure R5a** of this letter and **Figure 1** of the revised manuscript. We have also performed FRAP of α -Syn liquid droplets at 20 h to show that they are liquid-like (**Figure R5b** of this letter and **Figure 5** of the revised manuscript). Briefly, $100 \mu\text{M}$ α -Syn (spiked with $1 \mu\text{M}$ Alexa488 α -Syn) is phase separated in the presence of 20% (w/v) PEG6000 and subjected to fluorescence recovery after photobleaching (FRAP) experiments. The droplet is bleached at the center with a ROI radius of $4 \mu\text{m}$. A 488 nm bleaching laser is used at 100% power

and complete fluorescence recovery is recorded within 40 s. The normalized fluorescence recovery for both Ddx4n1 and α -Syn is calculated after correction for ROIs with passive bleaching and background fluorescence¹¹⁻¹³.

Figure R5: Ddx4n1 and α -Syn forms dynamic, liquid-like assemblies

a. (Left panel) 5 μ l of 100 μ M Ddx4n1 phase separated sample is immediately spotted on a coverslip at 20 °C and droplets are visualized with the help of a bright field microscope (Zeiss Axio vert. A1) under 40x magnification. Fusion events are shown as time-lapse snapshots for 3 s. The successful fusion events are marked with yellow arrowheads at 0 s and 2 s. **(Right panel)** Normalized fluorescence recovery of Ddx4n1 droplets is plotted for 20 s. Representative images before bleaching, just after bleaching (0 s), at 10 s and at 20 s are shown. **b.** Normalized fluorescence recovery of α -Syn droplet is plotted for 20 s. Representative images before bleaching, just after bleaching (0 s), at 20 s and at 40 s are shown. The experiments were performed twice with similar results.

4) Figure 1d, line 164 “Our data showed that ... concentrations of dilute phase and dense phase...” To my understanding the method can measure the concentration in the dilute phase (the critical concentration), not the concentration in the dense phase. I also found a bit difficult to read Fig. 1d. It is up to the authors but my suggestion would be to split this panel in two, or modify it.

Response: We apologize for this confusion. As the reviewer correctly points out, currently Capflex can only measure the dilute phase concentration (and not the dense phase concentration) of a phase separated sample. We intended to emphasize that the dilute phase concentration and the dilute phase equivalent protein concentration partitioned into the dense phase can be quantified by our method. The statement, ‘*Our data showed the absolute concentrations of dilute phase and dense phase do not change with total protein concentration, only their relative volume fractions.*’—is now replaced with ‘*Our data showed the concentrations of dilute phase does not change with total protein concentration; only the relative volume fractions of dilute and dense phase.*’ in the revised version of our manuscript. The assumption that it is the volume fraction and not the concentration of the dense phase that increases is based on the currently accepted standard view of LLPS, where movement parallel to the concentration axis leads to changes in relative volume fraction.

After the reviewer’s suggestion, we have now separated **Figure 1d** (**Figure R6** of this letter and **Figure 1** in the revised manuscript) into three sub-panels (Ddx4n1 alone, Ca^{+2} and PEG) which, to our view has improved the readability of the results.

Figure R6: Ddx4n1 dilute phase concentration in presence of additives promoting LLPS

Dilute phase concentrations for Ddx4n1 alone (*left*) and as a function of additives (Ca^{+2} (*middle*) and PEG3000 (*right*)) at 20 °C is shown. Ca^{+2} is reported in mM with 100 µM Ddx4n1; PEG3000 is reported in % (w/v) with 50 µM Ddx4n1 and, Ddx4n1 alone is reported in µM. The data represents mean \pm SD for n=2 independent experiments.

5) Effect of CaCl_2 : a control with MgCl_2 could be performed to show the specific effect of Ca.

Response: We thank the reviewer for this valuable suggestion. We have now measured dilute phase concentrations of 100 µM Ddx4n1 (spiked with 500 nM YFP-Ddx4n1) in the presence of 10 mM MgCl_2 (highest corresponding concentration that was used for CaCl_2) using Capflex. Interestingly, our results showed that at 15 °C, the dilute phase concentration of 100 µM Ddx4n1 was substantially reduced in presence of 10 mM MgCl_2 (**Figure R7** of this letter and **Supplementary Figure 5** of the revised manuscript). The effect of Mg^{+2} was very similar to Ca^{+2} (**Figure R7** of this letter and **Supplementary Figure 5** of the revised manuscript). Our data indicates that the observable decrease in the dilute phase concentration is not specific for Ca^{+2} , rather, could be a more generic effect of bivalent cation binding of Ddx4n1 modulating its LLPS behavior^{14,16}. On the other hand, this effect is clearly not explainable by simple Debye screening, as monovalent salts actually decrease the driving force for LLPS, whereas divalent salts apparently generally increase this driving force. These observations are now incorporated in the revised manuscript.

Figure R7: Effect of Mg^{+2} on the phase separation behavior of Ddx4n1

Capflex traces for 100 µM Ddx4n1 (with 500 nM YFP Ddx4n1) in the absence (*blue*) and in the presence of 10 mM Ca^{+2} (*red*) and Mg^{+2} (*black*) is shown. The sample tray temperature was maintained at 50 °C and the capillary temperature

was kept at 15 °C for this experiment. The inset represents relative decrease of the baseline concentration in the presence of both Ca^{+2} and Mg^{+2} . Representative Capflex traces are shown.

6) Figure 1 caption (and other Figures): “with similar observations”: this statement should be quantified or independent experiments should be shown.

Response: We apologize for not being clear. We had put this statement on the reproducibility of our research data in each of the figure legends, which is encouraged by *Nature communications* (<https://www.nature.com/ncomms/submit/guide-to-authors>). In the revised version, all reproducibility statements are provided for individual panels wherever applicable. As per guidelines, please also note that all the raw data (in replicates wherever applicable) presented in the manuscript are provided as MS Excel files termed as ‘*source data*’ files in the revised submission.

7) Size distribution: size distributions of droplets at different initial protein concentrations can be estimated by bulk microscopy images and (at least qualitatively) compared with the distribution of the fluorescence intensity (direct quantitative comparison may be hindered by sedimentation and wetting in the bulk assay).

Response: We thank the reviewer for this comment. We agree that the evolution in the droplet size with total Ddx4n1 concentration (as shown with Capflex) should also be validated using conventional microscopy. We have now analyzed 100, 116, 133 and 167 μM of phase separated Ddx4n1 (spiked with 1 μM YFP Ddx4n1) samples using a confocal fluorescence microscope (LMI-005-Leica Microsystems). Our observations were consistent with our Capflex results with respect to increase in droplet size and widening of the distribution as a function of total protein concentration (**Figure R8** of this letter and **Supplementary Figure 6** of the revised manuscript). As the reviewer correctly points out, we did face some difficulties in obtaining representative images without droplet sedimentation/wetting. We carefully performed depth (Z) scanning for all the images to verify the spherical nature of the droplets. Sedimented droplets were not taken into account for size distribution.

Figure R8: Evolution of droplet size distribution of Ddx4n1 with total protein concentration

Representative fluorescence microscopic images of Ddx4n1 droplets are shown for 100, 116, 133 and 167 μM total protein concentrations. The right panel shows the distribution of the droplet sizes for 100, 116, 133 and 167 μM total Ddx4n1 concentrations for $n=100$ liquid droplets per sample. The values represent individual droplet diameters (size) and the black line represents the median of the distribution.

8) Affinity Ddx4-ssDNA: Stoichiometry of binding to individual protein molecules shows that more than a single DNA molecule interacts with one molecule of Ddx4, with low affinity. However, highly sub-

stoichiometric concentrations of ssDNA dissolve the droplets. In the conclusions, this is explained with a higher affinity (one order of magnitude) of ssDNA for the droplet state. How was the affinity for the droplet phase calculated? Wouldn't a different affinity of ssDNA for continuous and dense phase lead to a different stoichiometry in the droplets with respect to the overall stoichiometry (in contrast with what has been measured?).

Response: We thank the reviewer for pointing out this somewhat imprecise formulation. When we speak of an affinity of the droplet state for the DNA, we do give the impression of having determined this affinity in detail, which is not the case. Our argument is simply supposed to highlight the finding that a deleterious effect on droplet stability is observed at concentrations approximately one order of magnitude below the precisely determined (by FIDA) affinity in the dilute regime between DNA and protein. Therefore, if one wanted to define an affinity of the droplet state for the DNA, it is likely to be approximately one order of magnitude below the dilute phase affinity. However, the reviewer is certainly correct in pointing out that the stoichiometry of interactions with the droplet phase is probably very different, because of the possibility to simultaneously interact with several protein molecules at the same time. We have re-formulated the corresponding section accordingly.

9) Fig. 3d: “possible transition from binodal to spinodal”: this statement is just briefly mentioned in the caption and is difficult to follow. More explanation is required in the main text.

Response: We apologize for the statement, ‘*The drop in the peak heights after 1.5 min represents possible transition from a binodal LLPS to a spinodal decomposition*’ which had been added by mistake, and which we had failed to remove in a revision prior to first submission.

10) Introduction: as current techniques to measure phase diagrams, NMR could also be mentioned although requiring large amount of sample (Brady J.P. et al., PNAS, 2017, 114, 39) and fluorescence correlation spectroscopy (Wei M.T. et al., Nature Chemistry, 1118-1125, 2017). For microfluidic approaches, in addition to ref. 52, please consider also Bremer A. et al., Lab Chip, 2020, 20, 4225-4234 and Kopp M. et al, 2020, 92, 8, 5803; instead or in addition to ref. 54 please consider Kueffner et al, ChemSystemsChem, 2020, 2, e2000001, where Raman spectroscopy was used earlier to measure protein concentration inside individual droplets.

Response: Thank you for suggesting the important references that we missed out. Please note that these references are now incorporated for appropriate statements in the revised version of our manuscript (reference no. 56, 51, 54, 55 and 59).

Reviewer #2 (Remarks to the Author):

The manuscript by Stender et al. entitled "Capillary flow experiments (Capflex) for thermodynamic and kinetic characterization of protein liquid-liquid phase separation at high throughput" describes a highly novel study that will make a significant impact in the field of liquid-liquid phase separation (LLPS). The authors utilize an automated Capillary Flow Experiment (abbreviated as Capflex) method that is based on Flow-Induced Dispersion Analysis (FIDA). This unique application of FIDA allowed the authors to obtain a wealth of highly quantitative information during LLPS of several unrelated intrinsically disordered proteins/peptides with or without nucleic acids. The authors demonstrated the power of FIDA-based Capflex in the quantitative estimation of the dilute phase concentration, droplet size distributions, and kinetics of LLPS and LLPS-mediated amyloid formation. Using this methodology, the authors were able to characterize the LLPS of Ddx4n1, complex coacervation of ssDNA and RP3, and LLPS/LLPS-mediated liquid-to-solid phase transition of alpha-synuclein into amyloids. This study represents a technological tour de force and is of great interest in quantitative in vitro characterizations of biological phase separation. This work is an important contribution and is accessible to a broad audience, and therefore, is suitable for publication in Nature Communications. Below I summarize my comments that can be addressed before this work is accepted for publication.

Response: We thank the reviewer for carefully reviewing our work and finding it interesting. The comments are of great importance to us, and will undoubtedly improve the quality of this work.

1. The authors state in lines # 163-165 "Our data showed the absolute concentrations of dilute phase and dense phase do not change with total protein concentration, only their relative volume fractions (Figure 1d)". They estimate the light phase concentration from the baseline fluorescence signal (using a standard curve calibration). How do they obtain the dense phase concentration? The dense phase concentration estimation might be difficult using this method because the peak intensities (or integral intensities) of the spikes (or bursts) are highly convoluted due to the dense phase concentration, a wide droplet size distribution, and bias due to the laminar flow (as the authors stated in lines 210-212: "rapid flow in the center of the capillary or slower flow at the edges"). Can the authors comment on the approximate dense phase concentration? I am not sure if some form of autocorrelation analysis can help determine the approximate dense phase concentration. If the dense phase concentration cannot be estimated by the Capflex method, I was wondering if the authors could consider stating this limitation.

Response: We apologize for this confusion. The reviewer is correct in that Capflex can only measure the dilute phase concentration (and not the dense phase concentration) of a phase separated sample. Therefore, the statement '*Our data showed the absolute concentrations of dilute phase and dense phase do not change with total protein concentration, only their relative volume fractions.*'—is now replaced with '*Our data showed the concentrations of dilute phase does not change with total protein concentration; only the relative volume fractions of dilute and dense phase.*' in the revised version of our manuscript. With the current FIDA1 instrument setup, calculation of dense phase concentration or volume fraction is challenging. The peak intensity corresponding to each droplet is a convolution of droplet size, dense phase concentration and also depends on the time a droplet stays in the detector window—which can be different depending on the position of the droplet inside the capillary flow (center or sides). We accept that it is a limitation of our study and have mentioned it clearly in the revised version of our manuscript. Our assumption that it is the volume fraction and not the concentration of the dense phase that increases is based on the currently accepted standard view of LLPS, where movement parallel to the concentration axis on the phase diagram leads to changes in relative volume fraction.

2. It's a bit hard to follow Figure 1d. Authors can consider simplifying this figure to clearly show the dependence on PEG, CaCl₂, etc. The unit is missing for the total protein concentration on the right ordinate (Y-axis).

Response: We thank reviewers #1 and #2 for this suggestion. We have now separated **Figure 1d (Figure #R1** of this letter and **Figure 1** in the revised manuscript) into three sub-panels (Ddx4n1 alone, Ca⁺² and PEG) which we agree has improved the readability of the results.

Figure #R1: Ddx4n1 dilute phase concentration in presence of additives promoting LLPS

Dilute phase concentrations for Ddx4n1 alone (*left*) and as a function of additives (Ca⁺² (*middle*) and PEG3000 (*right*)) at 20 °C is shown. Ca⁺² is reported in mM with 100 μM Ddx4n1; PEG3000 is reported in % (w/v) with 50 μM Ddx4n1 and, Ddx4n1 alone is reported in μM. The data represents mean ± SD for n=2 independent experiments.

3. Kinetics of Ddx4n1 LLPS. These results are very interesting (Figure 3). The authors directly show an acceleration of LLPS as a function % of PEG. Does the delay they observe at a lower % of PEG (2%) indicate a nucleation phenomenon? It's a bit confusing whether this 30 sec delay is partly due to the dead-time of their observation. The dead-time (~ 15 sec) is not clearly mentioned in the manuscript and stated in lines 307-308 and at the end in the discussion section (lines 549-550). The question is whether there is a nucleation process that exhibits sigmoidal kinetics or this delay (lag time) is related to the dead-time. Of course, in fast LLPS kinetic traces (3-5% PEG), this delay is shortened but not completely eliminated. And when the authors investigated pre-formed amyloid (LLPS-mediated alpha-synuclein amyloids) (Figure 5), they also observe ~2-3 min delay. It would be helpful for the readers if the authors could clarify the observed delays in their kinetic traces.

Response: We apologize for this confusion. The kinetic traces that are plotted in **Figure 3 (Figure 3** of the revised manuscript) of the manuscript do not show the instrument dead time. In the traces represented in the figure, the Ddx4n1 sample is injected into the capillary at highest pressure (3500 mbar) for 20 s until the entire capillary is filled. Subsequently, the kinetics of droplet formation is measured from time=0 at 20 °C capillary temperature with a reduced injection pressure (300 mbar). That is the reason why a sigmoidal curve is not observed in the traces. Indeed, the reviewer is correct in that the 30 s delay that is observed at lower PEG (2%) concentration could be due to a nucleation phenomenon during Ddx4n1 LLPS. However, with the current dataset, we can only propose that the appearance and subsequent increase in the peak heights represent droplet growth over the time-course of our experiment.

Notably, the dead time of the instrument depends on the sample viscosity and sample injection pressure and therefore, is not a constant measure. The extent of phase separation, which may lead to an increase in viscosity, also results in an increased dead time for Capflex for a given injection pressure. As the reviewer correctly

speculates, the delay in α -Syn peaks are only due to a longer dead time (at 1200 mbar injection pressure) since the samples are very viscous (20% (w/v) PEG6000). We have now demonstrated the dead time of our instrument for a sample having similar viscosity to water (1 nM Fluorescein in water). The calculated dead time was ~ 8 s at 3500 mbar of pressure. We have also explained the origin of the sigmoidal shape of Capflex traces schematically, which we believe will increase the readability of our manuscript (**Figure #R2** of this letter and **Figure 7** of the revised manuscript). Briefly, the sigmoidal pattern arises from the shape of the solution front passing the detector. In the parabolic Poiseuille flow profile of the FIDA1 capillary, the fluid in the center of the capillary flows faster than the fluid close to the walls. Due to this flow profile, the central part of the solution reaches the detector window first—leading to the sigmoidal increase in fluorescence signal (**Figure #R2** of this letter and **Figure 7** of the revised manuscript). We have now discussed this in the revised manuscript.

Figure #R2: Origin of sigmoidal traces in Capflex

1 nM Fluorescein in water is measured in Capflex at 3500 mbar injection pressure. The dead time of the instrument is the time it takes for the solution front to reach the detector window (black line of the scheme in the inset). The dead time of ~ 8 s is marked with a dashed red line in the actual data. The schematic depicts the progress of the solution front with time resulting in a sigmoidal increase of fluorescence, which reaches a stable baseline fluorescence after 12 s when the solution front has completely crossed the detector window.

4. Figure 3 legend (lines 341-342). "The drop in the peak heights after 1.5 min represents possible transition from a binodal LLPS to spinodal decomposition". This aspect is unclear and has not been explained in the manuscript text.

Response: We thank you and reviewer #1 very much for raising this concern. We apologize for the statement, *'The drop in the peak heights after 1.5 min represents possible transition from a binodal LLPS to a spinodal decomposition'* which had been added by mistake, and which we had failed to remove in a revision prior to first submission. We have now removed this statement.

Also, they state "The solid lines indicate the average increase in the peak intensity for respective conditions." It's not clear whether the solid lines are for guiding the eyes. The authors could consider stating this clearly.

Response: We apologize for not being clear regarding this. The solid lines are linear fits of the peak heights obtained from three independent observations for each sample (Ddx4n1 in presence of 2, 3 and 4% (w/v) PEG3000) from 0-2 min. The statement is now revised and incorporated in the revised figure legend (**Figure 3** of the revised manuscript).

5. Methods. Since this is primarily a Method manuscript and most readers may not be familiar with FIDA, I was wondering the authors could briefly state this methodology with some essential technical details including

dead time in a Methods section (after Discussion). Other experimental details can be in Supplementary Information.

Response: We thank the reviewer for this suggestion, which will undoubtedly improve our manuscript. The following paragraphs are now incorporated under the method section in the main manuscript;

The FIDA 1 consist of an autosampler holding two 96 well plates whose temperature can be individually controlled. The autosampler loads the sample directly into the 1 m long capillary that is housed in a separate thermostatted chamber. Three modes of operation are used in this study, standard Capflex (i.e. slow continuous injection of sample), rapid filling of the capillary followed by slow continued injection of the sample for the study of LLPS kinetics and standard flow-induced dispersion analysis (FIDA) for affinity determination. Notably, the dead time of the instrument (time taken for the solution front to reach the detector window) depends on the sample viscosity and sample injection pressure and therefore, is not a constant quantity.

1) For Capflex analysis, the capillary is equilibrated with sample buffer and the sample is then loaded into the capillary. The dead time here can be controlled by the applied pressure, but should preferably be long enough that a semi-equilibrium for LLPS has been achieved. The dilute phase concentration can then be determined from the fluorescence baseline. If the fluorescence signal stems from a small concentration of added labelled protein, the dilute phase concentration can be defined under the assumption that the labelled and unlabeled protein molecules partition into the droplets in the same manner. In future-versions of the instrument, it will also be possible to detect intrinsic fluorescence and hence unlabeled samples can be used.

2) For kinetic analysis the capillary is loaded at the highest possible pressure (3500 mbar) resulting in a dead time of ~8 s for a sample with viscosity similar to that of water. The measurement is then switched on, the pressure lowered and the kinetic development of the sample is followed.

For Capflex and kinetics of droplet formation, a sigmoidal pattern of the fluorescence intensity arises from the shape of the solution front passing the detector. The parabolic flow (i.e. Poiseuille) profile in the capillary, leads to the central part of the solution to move faster through the capillary than the solution closer to the capillary walls. Due to this flow profile, the center of the solution reaches the detector window first—leading to the observed sigmoidal profile (**Figure 7** of the revised manuscript).

3) For affinity determination¹⁷, the capillary is filled with a solution of interaction partner of known concentration. A small injection of 40 nl of the labeled biomolecule + interaction partner is then carried out and it is pushed through the capillary by the solution of interaction partner. The laminar flow in the capillary causes deformation of the plug, which is measured by a fluorescence detector. The degree of this deformation depends on the diffusivity of the labeled biomolecule. Hence, Diffusivity is directly measured which can be converted to hydrodynamic radius through the Stokes-Einstein relation. The change in hydrodynamic radius can then be fitted to the binding models described in the supplementary information.

The authors could consider citing an earlier/introductory paper on FIDA (J. Am. Chem. Soc. 2010, 132, 4070-4071 in which one of the authors of the current manuscript was an author).

Response: We thank the reviewer for pointing out to an important reference that we missed. The article is now referred in the revised version of our manuscript (**reference no. 80**).

6. Journal names are missing in some references (For example, Ref. 48-50).

Response: Journal names are incorporated for all the revised references. We thank the reviewer for pointing this out.

Reviewed by Samrat Mukhopadhyay.

Reviewer #3 (Remarks to the Author):

I currently have a split role in Industry and Academia. The work here is highly original and the ability to perform capillary high throughput analysis for quantitative LLPS is highly relevant and exciting to drug discovery activities in University and Industry. The multiple case studies presented amply support the conclusions and claims. I identified no flaws in the interpretation of this sound piece of work. The methods are provided in sufficient detail. I expect this approach will find wide interest to those working in drug discovery. I very much enjoyed reading the manuscript, which was easily understandable.

Response: We thank the reviewer for his/her positive evaluation of our work and we are particularly pleased to hear about the potential usefulness of our method in both an academic and industrial setting.

References:

- 1 Alberti, S., Gladfelter, A. & Mittag, T. Considerations and Challenges in Studying Liquid-Liquid Phase Separation and Biomolecular Condensates. *Cell* **176**, 419-434, doi:10.1016/j.cell.2018.12.035 (2019).
- 2 Wei, M.-T. *et al.* Phase behaviour of disordered proteins underlying low density and high permeability of liquid organelles. *Nature Chemistry* **9**, 1118-1125, doi:10.1038/nchem.2803 (2017).
- 3 Arter, W. E. *et al.* Rapid characterisation of protein phase behaviour using droplet microfluidics. 2020.2006.2004.132308 (2020).
- 4 Bremer, A., Mittag, T. & Heymann, M. Microfluidic characterization of macromolecular liquid-liquid phase separation. *Lab on a Chip* **20**, 4225-4234, doi:10.1039/D0LC00613K (2020).
- 5 Kopp, M. R. G. *et al.* Microfluidic Shrinking Droplet Concentrator for Analyte Detection and Phase Separation of Protein Solutions. *Analytical Chemistry* **92**, 5803-5812, doi:10.1021/acs.analchem.9b05329 (2020).
- 6 McCall, P. M. *et al.* Quantitative phase microscopy enables precise and efficient determination of biomolecular condensate composition. *bioRxiv*, 2020.2010.2025.352823, doi:10.1101/2020.10.25.352823 (2020).
- 7 Küffner, A. M. *et al.* Acceleration of an Enzymatic Reaction in Liquid Phase Separated Compartments Based on Intrinsically Disordered Protein Domains. **2**, e2000001, doi:<https://doi.org/10.1002/syst.202000001> (2020).
- 8 Brady, J. P. *et al.* Structural and hydrodynamic properties of an intrinsically disordered region of a germ cell-specific protein on phase separation. **114**, E8194-E8203, doi:10.1073/pnas.1706197114 %J Proceedings of the National Academy of Sciences (2017).
- 9 Milkovic, N. M. & Mittag, T. in *Intrinsically Disordered Proteins: Methods and Protocols* (eds Birthe B. Kragelund & Karen Skriver) 685-702 (Springer US, 2020).
- 10 Sawner, A. S. *et al.* Modulating α -Synuclein Liquid-Liquid Phase Separation. *Biochemistry*, doi:10.1021/acs.biochem.1c00434 (2021).
- 11 Ray, S. *et al.* α -Synuclein aggregation nucleates through liquid-liquid phase separation. *Nature Chemistry* **12**, 705-716, doi:10.1038/s41557-020-0465-9 (2020).
- 12 Axelrod, D., Koppel, D. E., Schlessinger, J., Elson, E. & Webb, W. W. Mobility measurement by analysis of fluorescence photobleaching recovery kinetics. *Biophys J* **16**, 1055-1069, doi:10.1016/S0006-3495(76)85755-4 (1976).
- 13 Soumpasis, D. M. Theoretical analysis of fluorescence photobleaching recovery experiments. *Biophys J* **41**, 95-97, doi:10.1016/S0006-3495(83)84410-5 (1983).
- 14 Crabtree, M. D. *et al.* Repulsive electrostatic interactions modulate dense and dilute phase properties of biomolecular condensates. (2020).
- 15 Nott, T. J., Craggs, T. D. & Baldwin, A. J. Membraneless organelles can melt nucleic acid duplexes and act as biomolecular filters. **8**, 569-575, doi:10.1038/nchem.2519 (2016).
- 16 Brady, J. P. *et al.* Structural and hydrodynamic properties of an intrinsically disordered region of a germ cell-specific protein on phase separation. **114**, E8194-E8203, doi:10.1073/pnas.1706197114 (2017).
- 17 Jensen, H. & Østergaard, J. Flow Induced Dispersion Analysis Quantifies Noncovalent Interactions in Nanoliter Samples. *Journal of the American Chemical Society* **132**, 4070-4071, doi:10.1021/ja100484d (2010).

Reviewers' Comments:

Reviewer #1:

Remarks to the Author:

The authors have done an outstanding job in fully addressing all my points. There is a large amount of new experiments and revision of the text which has further greatly improved the clarity and strength of the work. This work will be very valuable for the community.

Reviewer #2:

Remarks to the Author:

The authors have responded to all my comments and have revised their manuscript. The revised manuscript can now be accepted for publication.

Reviewer #1 (Remarks to the Author):

The authors have done an outstanding job in fully addressing all my points. There is a large amount of new experiments and revision of the text which has further greatly improved the clarity and strength of the work. This work will be very valuable for the community.

Response: We are grateful to the reviewer for critically reviewing our manuscript and help us improve the quality of the current study. We thank the reviewer for his/her positive comments and judging the revised manuscript suitable for publication in *Nature Communications*.

Reviewer #2 (Remarks to the Author):

The authors have responded to all my comments and have revised their manuscript. The revised manuscript can now be accepted for publication.

-Samrat Mukhopadhyay

Response: We thank the reviewer for helping us improve the quality of our manuscript. We thank the reviewer for judging the revised manuscript suitable for publication in *Nature Communications*.